# Differentiation signals induce *APOBEC3A* expression via GRHL3 in squamous epithelia and squamous cell carcinoma

Nicola J Smith [1,2,12], Ian Reddin [1,3,12], Paige Policelli [1,4], Sunwoo Oh[5], Nur Zainal [1], Emma Howes[1], Benjamin Jenkins[1], Ian Tracy[1], Mark Edmond[1], Benjamin Sharpe[1], Damian Amendra [1], Ke Zheng[6], Nagayasu Egawa[6], John Doorbar[6], Anjali Rao[7], Sangeetha Mahadevan[7], Michael A Carpenter[8,9], Reuben S Harris[8,9], Simak Ali [10], Christopher Hanley[1], Rémi Buisson[5], Emma King[1], Gareth J Thomas[1,11] & Tim R Fenton [1,11]✉

## Abstract

**Two APOBEC DNA cytosine deaminase enzymes, APOBEC3A and APOBEC3B, generate somatic mutations in cancer, thereby driving tumour development and drug resistance. Here, we used single-cell RNA sequencing to study *APOBEC3A* and *APOBEC3B* expression in healthy and malignant mucosal epithelia, validating key observations with immunohistochemistry, spatial transcriptomics and functional experiments. Whereas *APOBEC3B* is expressed in keratinocytes entering mitosis, we show that *APOBEC3A* expression is confined largely to terminally differentiating cells and requires grainyhead-like transcription factor 3 (GRHL3). Thus, in normal tissue, neither deaminase appears to be expressed at high levels during DNA replication, the cell-cycle stage associated with APOBEC-mediated mutagenesis. In contrast, in squamous cell carcinoma we find that, there is expansion of *GRHL3* expression and activity to a subset of cells undergoing DNA replication and concomitant extension of *APOBEC3A* expression to proliferating cells. These findings suggest that APOBEC3A may play a functional role during keratinocyte differentiation, and offer a mechanism for acquisition of APOBEC3A mutagenic activity in tumours.**

**Keywords** APOBEC3A; Cancer Mutagenesis; GRHL3; Keratinocyte; HNSCC
**Subject Categories** Cancer; Skin

## Introduction

The *APOBEC3A* and *APOBEC3B* (apolipoprotein B mRNA editing catalytic polypeptide-like 3A and 3B) genes encode two closely related DNA cytosine deaminases that belong to the seven-protein human APOBEC3 family. The APOBEC3 enzymes convert deoxycytidine to deoxyuridine in single-stranded DNA (ssDNA), a mutagenic activity that explains at least in part their ability to restrict replication of retroviruses and endogenous retroelements through targeting nascent cDNA during reverse transcription (Harris and Dudley, 2015; Conticello, 2008). In addition, APOBEC3A and APOBEC3B have evolved functions in the cell nucleus, including transcriptional regulation (Periyasamy et al, 2015; Taura et al, 2022) and responses to nuclear-resident viruses (Chen et al, 2006; Vartanian et al, 2008; Warren et al, 2015; Lejeune et al, 2023; Lucifora et al, 2014; Cheng et al, 2018). Acquisition of these nuclear functions appears to have come at a cost, however, as both APOBEC3A and APOBEC3B have been implicated in generating somatic mutations (mainly C > T transitions and C > G transversions at TpC sites) in cancer cell genomes, driving cancer development and therapeutic resistance (Henderson and Fenton, 2015; Swanton et al, 2015; Petljak and Maciejowski, 2020; Mertz et al, 2022; Green and Weitzman, 2019; Isozaki et al, 2023; Lin et al, 2022; Caswell et al, 2023; Periyasamy et al, 2020; Law et al, 2020; Durfee et al, 2023; Naumann et al, 2023). Two mutation signatures attributed to APOBEC3A/B activity have been observed in multiple cancer types, but although extensive biochemical and genetic data support the involvement of both APOBEC3A and APOBEC3B in somatic mutagenesis, their gene expression levels as determined by analysis of bulk tumour data or in cancer cell lines that display the signatures are only weakly, if at all, correlated with the presence of these mutation signatures (Alexandrov et al, 2013; Roberts et al, 2013;

[1]School of Cancer Sciences, Faculty of Medicine, University of Southampton, Southampton, UK. [2]School of Biosciences, University of Kent, Canterbury, UK. [3]Bio-R Bioinformatics Research Facility, Faculty of Medicine, University of Southampton, Southampton, UK. [4]Cell, Gene and RNA Therapies, Discovery Sciences, BioPharmaceuticals R&D, AstraZeneca, Cambridge, UK. [5]Department of Biological Chemistry, School of Medicine, University of California Irvine, Irvine, CA, USA. [6]Department of Pathology, University of Cambridge, Cambridge, UK. [7]Gilead Sciences, Research Department, 324 Lakeside Dr, Foster City, CA 94404, USA. [8]Department of Biochemistry and Structural Biology, University of Texas Health San Antonio, San Antonio, TX 78229, USA. [9]Howard Hughes Medical Institute, University of Texas Health San Antonio, San Antonio, TX 78229, USA. [10]Department of Surgery and Cancer, Imperial College London, Hammersmith Hospital Campus, London, UK. [11]Institute for Life Sciences, University of Southampton, Southampton, UK. [12]These authors contributed equally: Nicola J Smith, Ian Reddin. ✉E-mail: t.r.fenton@soton.ac.uk

Burns et al, 2013b, 2013a; Taylor et al, 2013; Cortez et al, 2019; Buisson et al, 2019; Jalili et al, 2020; Petljak et al, 2022a). Accurately determining the conditions under which *APOBEC3A* and *APOBEC3B* are expressed in normal and cancerous tissues represents a key challenge in building our understanding of how APOBEC-mediated mutagenesis occurs, and how they might be targeted for cancer treatment (Petljak et al, 2022b). This objective is complicated, however, by their expression in immune cells, which are frequently present at high levels in tumour biopsies, and until recently by a lack of specific antibodies for in situ analysis. Here, we addressed these challenges by conducting single-cell RNA sequencing (scRNA-seq) of matched normal and tumour samples from patients with head and neck squamous cell carcinoma (HNSCC), a tumour type in which high burdens of APOBEC signature mutations are frequently observed, with evidence pointing to roles for both APOBEC3A and APOBEC3B in generating these mutations (Alexandrov et al, 2013; Burns et al, 2013b; Roberts et al, 2013; Henderson et al, 2014; The Cancer Genome Atlas Network, 2015; Faden et al, 2017). We analysed *APOBEC3A* and *APOBEC3B* gene expression patterns in these data and in additional published scRNA-seq datasets from healthy and cancerous epithelial tissues, deploying recently developed antibodies in immunohistochemical analysis of tissue sections to corroborate our findings at the protein level. We used computational methods to predict transcription factors responsible for regulating APOBEC expression and validated our predictions in near-normal immortalised keratinocytes (NIKS) (Allen-Hoffmann et al, 2000), identifying Grainyhead-like transcription factor 3 (GRHL3) as a novel regulator of *APOBEC3A* expression in terminally differentiating keratinocytes. In contrast, and consistent with findings from different cell types (Hirabayashi et al, 2021; Roelofs et al, 2023), *APOBEC3B* expression is confined to proliferating cells, with the highest levels evident in the G2/M-phase of the cell cycle. In HNSCC, we find evidence of GRHL3 activity and *APOBEC3A* in a subpopulation of tumour cells undergoing DNA replication; the context in which mutagenic APOBEC activity is postulated to occur due to deamination of lagging strand ssDNA exposed at the replication fork (Green et al, 2016; Haradhvala et al, 2016; Hoopes et al, 2016; Morganella et al, 2016; Seplyarskiy et al, 2016; Stewart et al, 2020). Our findings provide new insight into the transcriptional control of *APOBEC3A* gene expression in squamous epithelia and provide a potential mechanism for the acquisition of APOBEC3A-induced mutations in cancer.

# Results

## APOBEC3A is expressed in epithelial cells from healthy tonsil and oesophagus

Although most cancers that display enrichment for APOBEC mutational signatures are carcinomas, i.e. tumours that arise from epithelial cells, little is known about the expression patterns and physiological regulation of *APOBEC3A* and *APOBEC3B* in healthy epithelium, or regarding the proportion of malignant cells that express *APOBEC3A*, *APOBEC3B* or both genes in tumour biopsies. To address these knowledge gaps, we assembled scRNA-seq data from the epithelial cells (see Methods) present in ten oropharyngeal SCC samples and seven matched normal (contralateral tonsil) samples from patients undergoing surgical resection at our institution (Table EV1), together with 11 published scRNA-seq datasets from healthy skin, breast and oesophagus and from cancers of the breast,

bladder, head and neck (HNSCC), oesophagus (ESCC) and lung (all cancers that typically display moderate to strong enrichment for APOBEC mutation signatures, Table EV2). Very few *APOBEC3A* or *APOBEC3B*-expressing epithelial cells were present in the normal skin, breast or lung datasets, but 16.4% of epithelial cells from normal oesophagus and 38.1% from normal tonsil expressed *APOBEC3A*, 10.5% of which also expressed *APOBEC3B* (Fig. 1). 3.5% of tonsil epithelial cells expressed only *APOBEC3B*, and the mean *APOBEC3A* expression per cell was significantly higher than that of *APOBEC3B* in both tonsil and eosophagus (Fig. 1, $p < 2.22E\text{-}16$ (unpaired Wilcoxon's rank-sum test)). In the tumour samples, the majority of epithelial (tumour) cells expressed neither *APOBEC3A* nor *APOBEC3B* at levels detectable by scRNA-seq, and the only datasets containing a significant number of cells expressing *APOBEC3A* and/or *APOBEC3B* were from HNSCC or ESCC (Fig. 1). Only the datasets from healthy tonsil and oesophagus and from HNSCC and ESCC contained sufficient *APOBEC3A* and/or *APOBEC3B* expressing cells to permit further analysis, so we initially interrogated the data from tonsil epithelial cells, the dataset in which we observed the highest average *APOBEC3A* expression per cell and the greatest proportion of *APOBEC3A*- and/or *APOBEC3B*-positive cells (Fig. 1).

## APOBEC3A and APOBEC3B are expressed in distinct cell sub-populations in healthy tonsil epithelium

Since most *APOBEC3A*-expressing cells in the normal tonsil dataset did not co-express detectable levels of *APOBEC3B*, we were interested in determining whether the cells comprising this *APOBEC3A*-positive/*APOBEC3B*-negative population might share a common phenotype and if so, whether it might be distinct from the *APOBEC3A*-negative/*APOBEC3B*-positive, *APOBEC3A/B*-positive and *APOBEC3A/B*-negative populations. To address this question, after further quality control (see Methods for details), we identified the top 100 genes co-expressed with either *APOBEC3A* or *APOBEC3B* (Table EV3) in 2649 epithelial cells from our healthy tonsil samples (Fig. 2A, blue) and used pathway analysis to examine the gene ontology biological processes (GOBP) enriched among each geneset (Fig. 2B). Considering only the top ten GOBP pathway hits for the two APOBECs, there was no overlap, and each was dominated by different biological processes. For *APOBEC3A*, the top ten pathways included those involved in epidermal and keratinocyte development and differentiation, whereas, consistent with observations in bulk RNA-seq data from breast cancer (Cescon et al, 2015), all processes in the top ten for *APOBEC3B* were associated with mitosis (Fig. 2B; Dataset EV1). This finding suggested that in healthy tonsil epithelium, *APOBEC3B* is expressed in cycling cells undergoing cell division while *APOBEC3A* is restricted to those keratinocytes undergoing terminal differentiation. To further investigate this possibility, the epithelial cells were clustered based on known markers for different epithelial cell states (Kang et al, 2015; Kabir et al, 2022; Rochman et al, 2022; Franzén et al, 2019). These genes included markers for basal cells (cytokeratin-14 (KRT14) and cytokeratin-15 (KRT15)), proliferating epithelium (Ki-67 (MKI67), minichromosome maintenance complex component 7 (MCM7)), differentiating keratinocytes (involucrin (IVL), cytokeratin-10 (KRT10)), and terminally differentiating keratinocytes (SPRR2A, S100P) (Fig. 2C,D). Similarly, scRNA-seq of the healthy oesophageal epithelium (Madissoon et al, 2019) had previously been clustered into four distinct epithelial phenotypes: basal epithelium ('epi-basal'), a proliferating suprabasal epithelium ('epi-suprabasal'), differentiating stratified epithelium

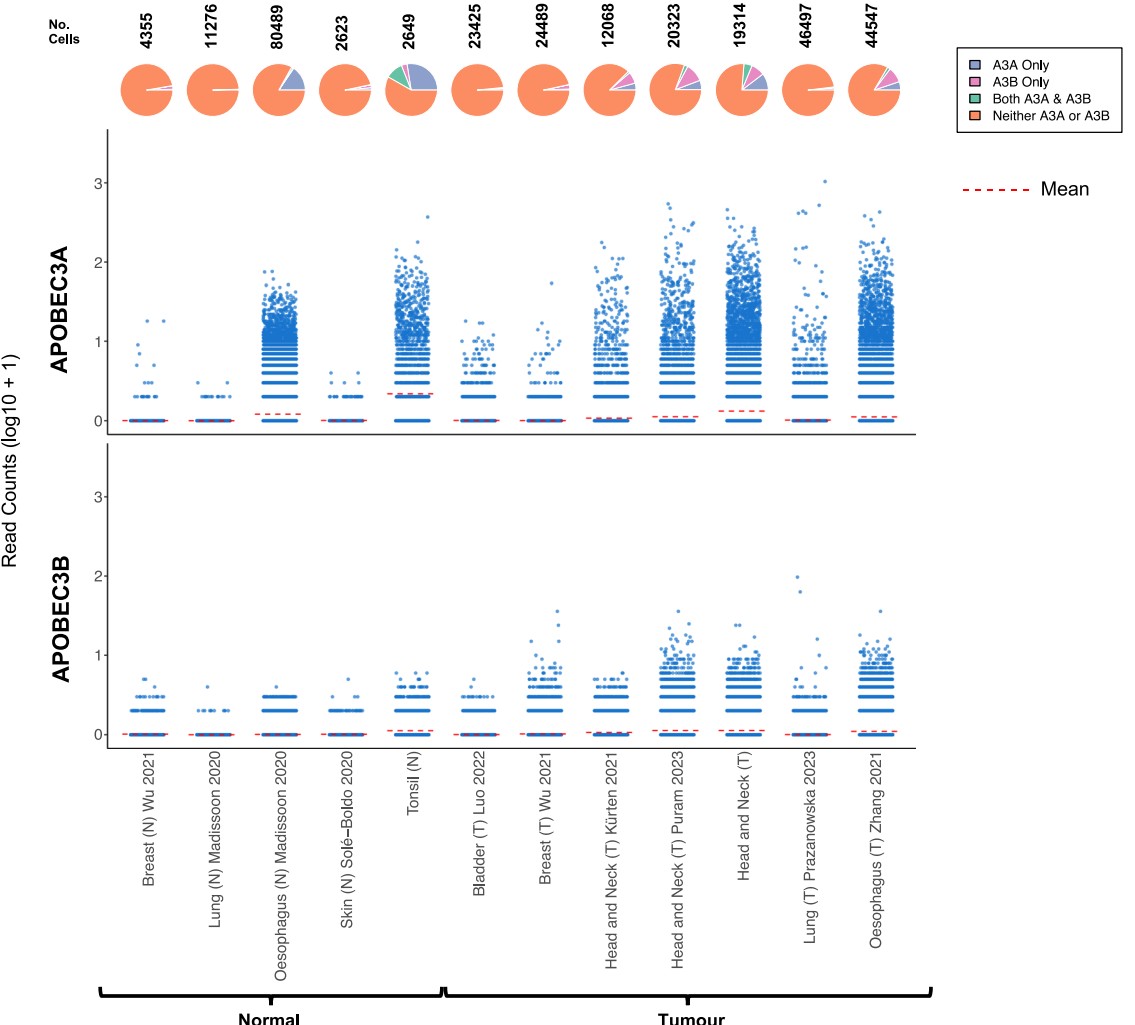

**Figure 1.  *APOBEC3A* and *APOBEC3B* expression in scRNA-seq datasets representing normal and tumour epithelial cells from tissues in which cancers that display prominent APOBEC mutational signatures arise.**

The number above each pie chart represents the total number of epithelial cells in each dataset. The references for each dataset are provided in Table EV2.

('epi-stratified'), and terminally differentiated upper epithelium ('epi-upper') (Appendix Fig. S1A). Marker gene expression patterns for these epithelial subtypes resembled those observed in the corresponding tonsillar epithelial subtypes (Appendix Fig. S1B).

As inferred from the pathway analysis, *APOBEC3B* was expressed predominantly in proliferating cells, significantly more so than in differentiating cells ($p < 0.0001$, Wilcoxon's rank-sum test), exhibiting a similar expression profile to *MKI67* (Fig. 2E). While also expressed in a subset of proliferating cells, *APOBEC3A* expression was significantly higher in differentiating cells ($p < 0.0001$, Wilcoxon's rank-sum test), and was also detectable in some terminally differentiated cells, following the expression pattern of *IVL* (Fig. 2E). Although *APOBEC3A* was expressed in a lower proportion of healthy oesophageal epithelial cells compared to those of the tonsillar epithelium, it was again co-expressed with *IVL* in differentiating cells and expressed weakly if at all, in the proliferative compartment ($p < 0.0001$, Wilcoxon's rank-sum test, Appendix Fig. S1C).

## Keratinocyte cell cycle exit and initiation of differentiation is marked by a switch from *APOBEC3B* to *APOBEC3A* expression

Our finding that *APOBEC3A* is expressed in differentiating epithelial cells of the tonsil and oesophagus is consistent with a previous report that it is upregulated during $Ca^{2+}$-induced differentiation of W12 cells (a cell line established from cervical neoplasia that harbours HPV16 (Stanley et al, 1989)) and normal human epidermal keratinocytes (NHEK) (Wakae et al, 2018). In the same study, *APOBEC3B* upregulation was also observed in W12 cells following 10 days in high $Ca^{2+}$ but not in NHEKs, suggesting an HPV-specific induction, as reported (Vieira et al, 2014; Henderson et al, 2014; Periyasamy et al, 2017). $Ca^{2+}$ is an activator of protein kinase C (PKC) signalling, which also mediates potent induction of *APOBEC3A* by phorbol esters in keratinocytes (Rasmussen and Celis, 1993; Madsen et al, 1999; Siriwardena et al, 2018). We therefore sought to uncouple potential PKC-dependent effects from differentiation-dependent effects on

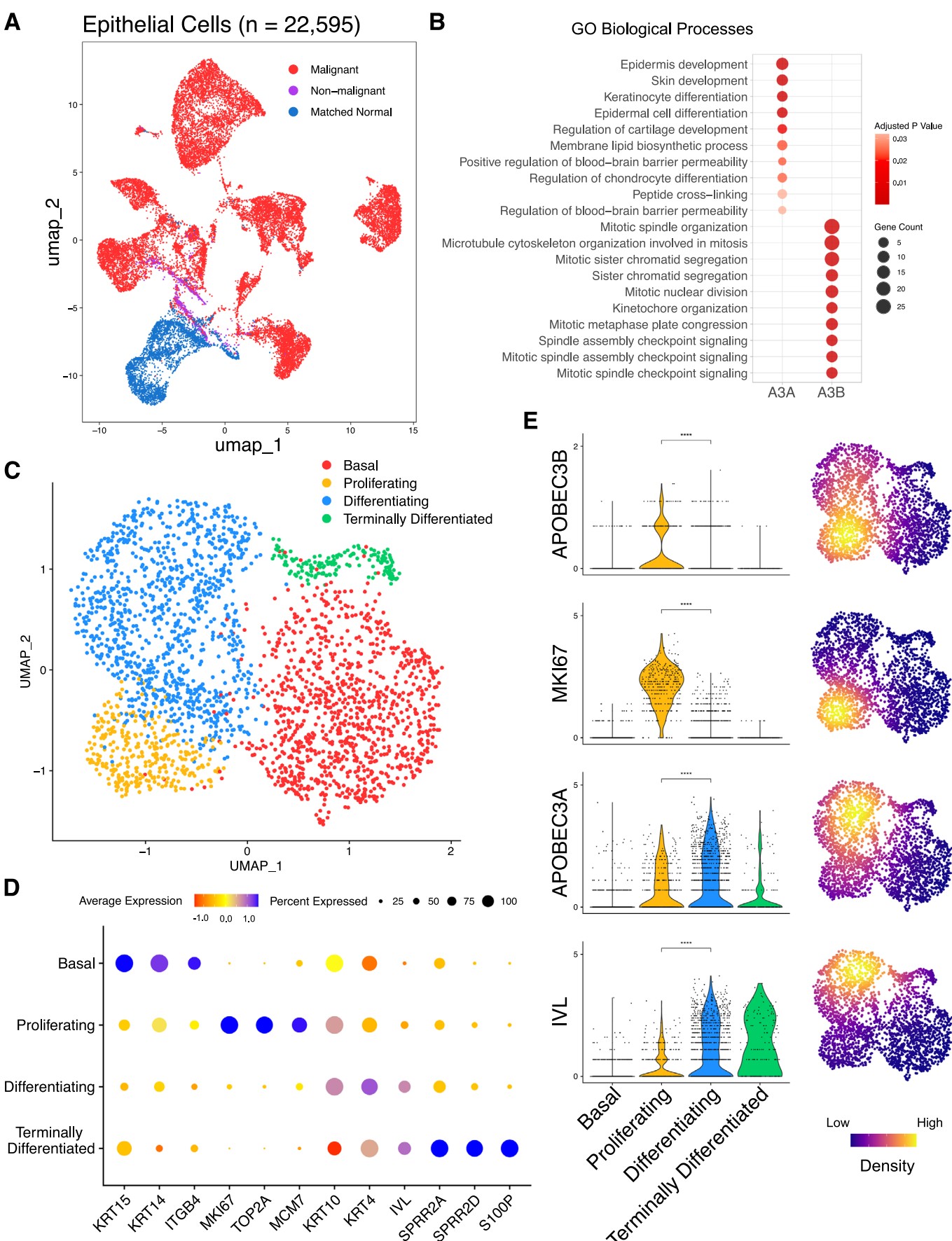

**A** Epithelial Cells (n = 22,595)

**B** GO Biological Processes

**C**

**D**

**E**

**Figure 2. *APOBEC3A* and *APOBEC3B* are expressed in different subsets of tonsillar epithelial cells.**

(A) UMAP projection of 22,595 epithelial cells from oropharyngeal squamous cell carcinoma samples ($n = 10$; 18,619 malignant cells (red), 695 non-malignant cells (purple)), and matched normal tonsil ($n = 7$; 3281 cells (blue)). (B) GOBP terms enriched amongst the sets of 100 genes that were co-expressed with either *APOBEC3A* or *APOBEC3B* in 2649 (after QC) epithelial cells from normal tonsil. Terms ranked by $P$ value calculated by EnrichR package (Fisher Exact test bias corrected using the z-score of the deviation from the expected rank). (C) UMAP projection depicting four phenotypes (basal, proliferating, differentiating, terminally differentiated) displayed by the normal tonsillar epithelial cells in our dataset. (D) Marker genes are used to identify the four epithelial phenotypes represented in (C). (E) Violin plots of gene expression in individual tonsillar epithelial cells, and UMAP projections of the density of gene expression in the tonsillar epithelial subtypes. (****$P < 0.0001$, Wilcoxon's rank-sum test; Basal, $n = 1047$; Proliferating, $n = 353$; Differentiating, $n = 1091$; Terminally differentiated, $n = 158$).

*APOBEC3A* expression by using three other established methods for inducing keratinocyte differentiation: growth to high density; inhibition of epidermal growth factor receptor (EGFR); and growth factor withdrawal (Poumay and Pittelkow, 1995; Peus et al, 1997). In all three contexts, we observed upregulation of *APOBEC3A*, along with *IVL* and *KRT10*, and either no change (following 24 h of the EGFR inhibitor afatinib) or a decrease (following growth to high density, serum and growth factor withdrawal) in *APOBEC3B* expression, closely mirroring decreases in *MKI67* and *MCM7* (Fig. 3A,C,E). Analysis of DNA content by flow cytometry confirmed cell cycle arrest in all three conditions (Fig. 3B,D,F). In contrast to *APOBEC3A* induction by phorbol 12-myristate 13-acetate (PMA), the upregulation observed following growth factor withdrawal was independent of PKC signalling (Appendix Fig. S2A,B) and was also observed in primary keratinocytes (Fig. 3G). Editing of cytosine 558 to uracil in the Dolichyl-diphosphooligosaccharide-protein glycosyltransferase 48 kDa subunit (DDOST) transcript (a highly specific measure of APOBEC3A deaminase activity (Jalili et al, 2020)) paralleled the increase in APOBEC3A mRNA observed upon growth factor withdrawal (Fig. 3H). Upon treating NIKS with inhibitors of the two major mitogenic signalling pathways downstream of EGFR (RAS/MEK/ERK and PI3K/AKT/mTOR) we observed induction of *APOBEC3A* only by the MEK inhibitor trametinib, which was also the only compound to induce *IVL/KRT10* expression (Appendix Fig. S2C, top) and to induce cell cycle arrest (Appendix Fig. S2D). Interestingly the ERK1/2 inhibitor ravoxertinib did not block proliferation, nor did it induce *APOBEC3A* or differentiation markers (Appendix Fig. S2C,D). PI3K (pictilosib), AKT (MK2206) and mTORC1 (everolimus) inhibitors had no effect on proliferation or *APOBEC3A* expression, but they did reduce *APOBEC3B* expression, which, unlike *MKI67* and *MCM7*, was unaffected by MEK inhibition (Appendix Fig. S2C). PI3K inhibition has previously been shown to reduce *APOBEC3B* expression in the U2OS human osteosarcoma cell line, via effects on NFκB and AP-1 activity (Lin et al, 2020).

Taken together, our results from human tissue samples and the experiments in NIKS suggest that cell cycle exit and initiation of terminal differentiation in keratinocytes is accompanied by a switch in *APOBEC3* gene usage, from *APOBEC3B*, which is expressed in cycling cells entering cell division, to *APOBEC3A*.

## APOBEC3A expression during keratinocyte differentiation requires Grainyhead-like transcription factor 3

Transcription factor activity analysis of scRNA-seq data from normal tonsil epithelial cells using single-cell regulatory network inference and clustering (SCENIC) (Aibar et al, 2017) identified the Grainyhead-like transcription factor 3 (GRHL3), a key regulator of

epidermal differentiation (Ting et al, 2005; Yu et al, 2006; Hopkin et al, 2012; Klein et al, 2017) as a potential regulator of *APOBEC3A* expression in the datasets from normal tonsil, HNSCC and ESCC, with strong positive associations between GRHL3 activity scores and *APOBEC3A* expression evident across all scRNA-seq datasets we analysed (Dataset EV2). Furthermore, GRHL3 was the only transcription factor among those whose activity was correlated with *APOBEC3A* expression, and significantly increased in the differentiating compartment of the normal tonsil epithelium, in which most *APOBEC3A*-expressing cells were clustered (Figs. 4A,B, 2E; Table EV4). It is also known to be activated downstream of the receptor-interacting protein kinase 4 (RIPK4) in PMA-treated keratinocytes (Scholz et al, 2016). Stratifying cells by their binary (on/off) GRHL3 activity as determined from SCENIC analysis (Appendix Fig. S3) revealed increased *APOBEC3A* expression in 'GRHL3-on' cells (Fig. 4C (upper panel); Wilcoxon's rank-sum test $p < 0.0001$). 936 of 1416 (66%) 'GRHL3-on' cells expressed *APOBEC3A*, a significantly higher number of cells compared to those that were 'GRHL3-off', where only 73 of 1233 (6%) cells expressed *APOBEC3A* (Fishers Exact Test, $p < 0.0001$; Fig. 4C (lower panel)). Published GRHL3 target genes include *IVL* and E74 like ETS transcription Ffactor 3 (*ELF3*) (Scholz et al, 2016; Hopkin et al, 2012), both of which display very similar patterns of gene expression to *APOBEC3A* in response to differentiation stimuli in NIKS (Fig. 3; Appendix Fig S2 and Appendix Fig S4). Suppressing *GRHL3* expression using two different siRNAs blocked the induction of *APOBEC3A* mRNA (Fig. 4D) and DDOST mRNA editing (Fig. 4E) by afatinib in NIKS, demonstrating a functional role for GRHL3 in activating *APOBEC3A* expression during differentiation. Induction of GRHL3 target genes *IVL* and *ELF3* was also suppressed, whereas expression of *MKI67* and *MCM7* was unaffected by GHRL3 knockdown (Appendix Fig. S5), consistent with GHRL3-dependent induction of *APOBEC3A* occurring during afatinib-induced differentiation, downstream of cell cycle exit. *GRHL3* knockdown also decreased *APOBEC3A* mRNA levels in PMA-treated NIKS (Appendix Fig. S6A), and following growth factor withdrawal (Appendix Fig. S6B,C). Analysis of chromatin immunoprecipitation (ChIP-seq) data from human keratinocytes (NHEK) (Hopkin et al, 2012; Klein et al, 2017) revealed GRHL3 binding at a predicted enhancer 33 kb upstream of the *APOBEC3A* TSS following $Ca^{2+}$-induced differentiation but not in control (proliferating) cells, consistent with a direct role for GRHL3 in regulating *APOBEC3A* transcription (Fig. 4F; Appendix Fig. 7A, main panel). The 176 bp region at which the GRHL3 binding peak is located contains four 8-mer sequences that are close matches for the previously defined consensus GRHL3 binding motif (AACC[G/T]GTT) (Yu et al, 2006) (Appendix Fig. 7A inset). GRHL3 has been shown to recruit the trithorax group (trxG) protein WDR5 to its

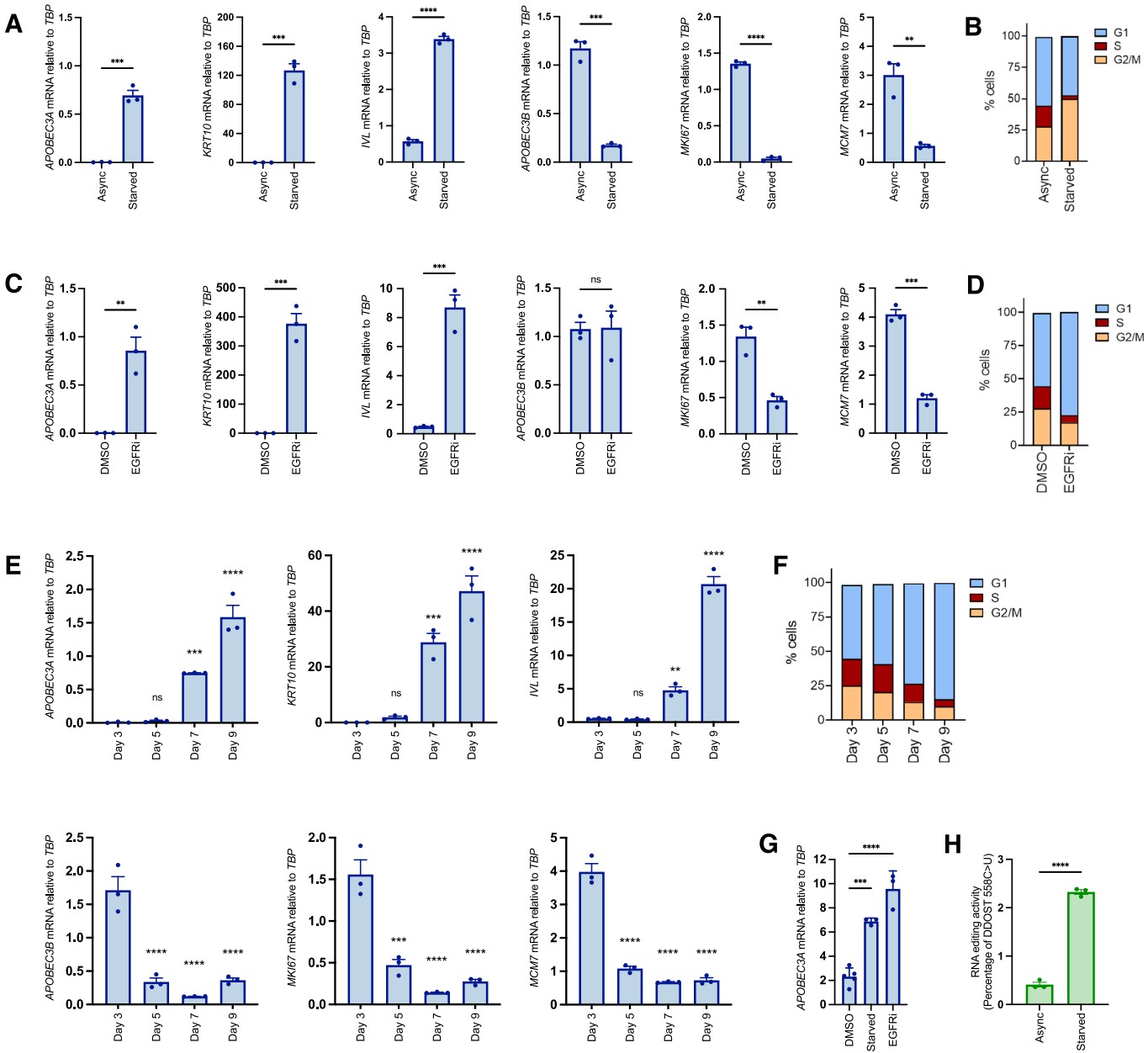

**Figure 3.  Keratinocyte cell cycle exit and initiation of differentiation is marked by a switch from *APOBEC3B* to *APOBEC3A* expression.**

(A) qRT-PCR-based gene expression measurements ($n = 3$) for *APOBEC3A* ($P = 0.0002$), *KRT10* ($P = 0.0002$), *IVL* ($P < 0.0001$), *APOBEC3B* ($P = 0.0001$), *MKI67* ($P < 0.0001$) and *MCM7* ($P = 0.0033$) in proliferating NIKS (Async) or following 48 h of growth factor deprivation (Starved). (B) Representative cell cycle profiles of Async and starved NIKS measured by PI staining and flow cytometry. (C) qRT-PCR-based gene expression measurements ($n = 3$) for *APOBEC3A* ($P = 0.0036$), *KRT10* ($P = 0.0004$), *IVL* ($P = 0.0007$), *APOBEC3B* ($P = 0.9445$), *MKI67* ($P = 0.0032$) and *MCM7* ($P = 0.0001$) following 24 h of vehicle control (DMSO) or 100 nM afatinib treatment (EGFRi). (D) Representative cell cycle profiles of DMSO and afatinib-treated NIKS measured by PI staining and flow cytometry. (E) qRT-PCR-based gene expression measurements ($n = 3$) for *APOBEC3A* (adjusted $P$ values versus day 3 measurement: day 5 = 0.9960; day 7 = 0.0009; day 9 <0.0001), *KRT10* (adjusted $P$ values versus day 3 measurement: day 5 = 0.9518; day 7 = 0.0005; day 9 <0.0001), *IVL* (adjusted $P$ values versus day 3 measurement: day 5 = 0.9983; day 7 = 0.0035; day 9 <0.0001), *APOBEC3B* (adjusted $P$ values versus day 3 measurement: day 5 <0.0001; day 7 <0.0001; day 9 <0.0001), *MKI67* (adjusted $P$ values versus day 3 measurement: day 5 = 0.0001; day 7 <0.0001; day 9 <0.0001) and *MCM7* (adjusted $P$ values versus day 3 measurement: day 5 <0.0001; day 7 <0.0001; day 9 <0.0001) in NIKS collected 3, 5, 7, or 9 days after plating. (F) Representative cell cycle profiles of NIKS collected 3, 5, 7, or 9 days after plating were measured by PI staining and flow cytometry. (G) qRT-PCR measurements of *APOBEC3A* expression in primary human epidermal keratinocytes (NHEK) growing in FC medium and treated for 24 h with vehicle control (DMSO) or 100 nM afatinib (EGFRi, $P < 0.0001$) or following growth factor deprivation for 48 h (Starved, $P = 0.0003$). (H) Percentage of DDOST transcripts that were C > U edited at c558 in asynchronous growing NIKS (Async) and following 48 h of growth factor withdrawal (starved) measured by digital PCR assay ($n = 3$; $P < 0.0001$). Error bars = SEM. *$P < 0.05$; **$P < 0.01$; ***$P < 0.001$; ****$P < 0.0001$. Pairwise comparisons were performed using unpaired two-tailed *t*-tests in (A, C, H). Comparisons of mRNA levels on days 5, 7 and 9 to day 3 in (E), and of mRNA levels in starved and afatinib-treated cells to DMSO-treated cells in (G) were performed using one-way ANOVA with Dunnett's multiple comparisons test. Source data are available online for this figure.

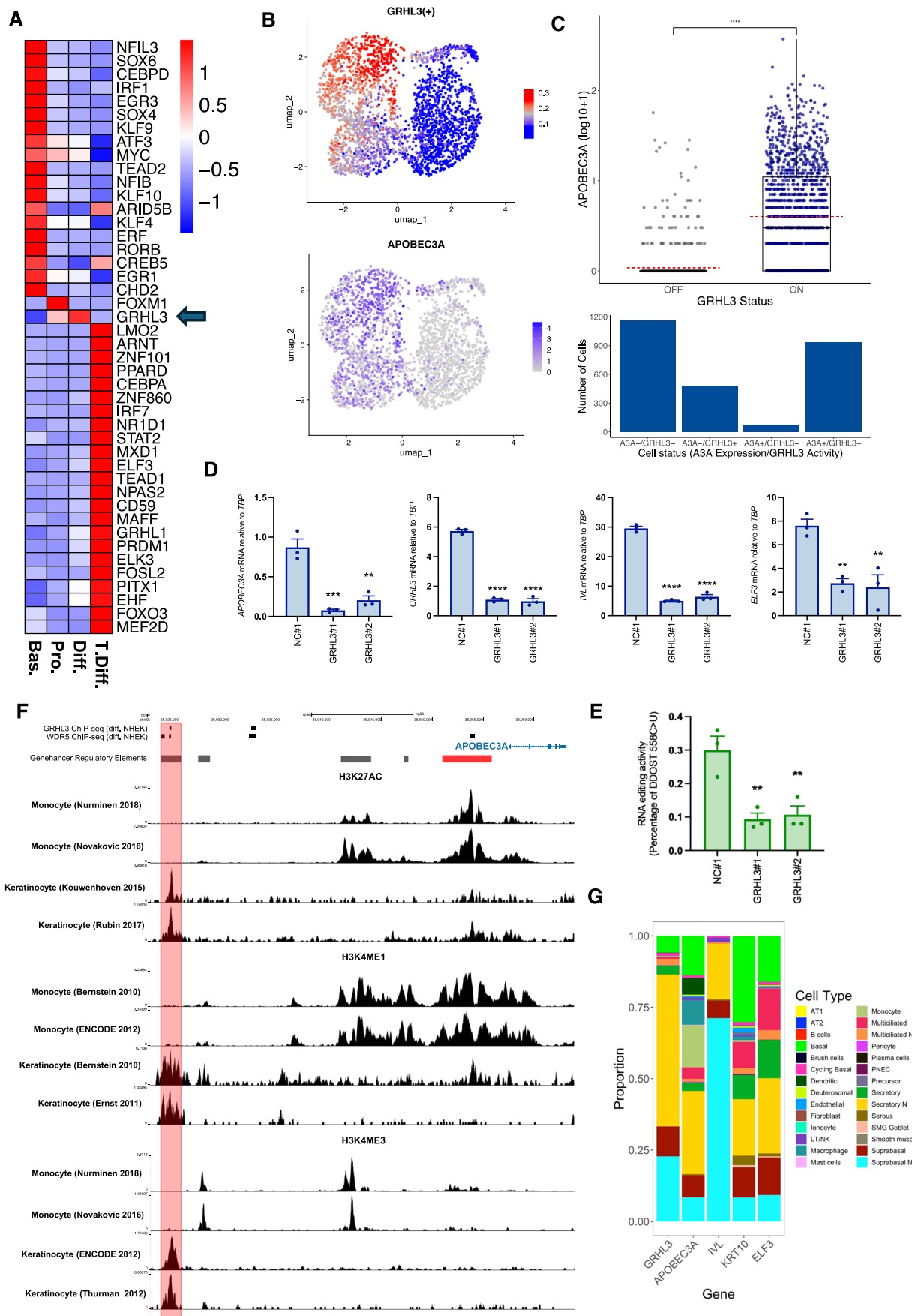

◄

**Figure 4.  Grainyhead-like transcription factor 3 is required for *APOBEC3A* expression during keratinocyte differentiation.**

(A) Heatmap showing those transcription factors (of the 363 with a SCENIC activity score in our scRNA-seq dataset from healthy tonsil epithelium) that were differentially active (fold change >1.1, adjusted $P < 0.05$, Wilcoxon rank-sum test) between the clusters defined in Fig. 2C (Bas. basal; Pro. proliferating; Diff. differentiating; T.Diff. terminally differentiated). (B) UMAPs showing GRHL3 transcription factor activity score from SCENIC (top) and *APOBEC3A* expression (bottom) in the Southampton scRNA-seq dataset from healthy tonsil epithelium. (C) Boxplot showing *APOBEC3A* expression stratified by SCENIC binary predictions of GRHL3 'off' or GRHL3 'on' (top; (****$P < 0.0001$, Wilcoxon's rank-sum test)) and histogram showing the number of cells in each of four groups: GRHL3 'off', no detectable *APOBEC3A* (A3A-/GRHL3-), $n = 1,160$; GRHL3 'on', no detectable *APOBEC3A* (A3A-/GRHL3+), $n = 480$; GRHL3 'off', *APOBEC3A* expressed (A3A+/GRHL−), $n = 73$, and GRHL3 'on', *APOBEC3A* expressed (A3A+/GRHL3+), $n = 936$ (bottom). The top, middle and bottom lines of the boxplot represent the upper quartile (Q3), median, and lower quartile (Q1), respectively. The maximum value represented by the top whisker represents the highest observed data point within Q3 + (1.5 × (Q3 − Q1)), and the minimum value represented by the bottom whisker represents the lowest situated point within Q1 − (1.5 × (Q3 − Q1)). The dashed red line represents the mean. (D) qRT-PCR-based expression measurements ($n = 3$) of *APOBEC3A, GRHL3, IVL* and *ELF3* in NIKS transfected with control (NC#1) or *GRHL3*-specific siRNAs as indicated. Adjusted $P$ values for comparisons between NC#1 and *GRHL3* siRNA #1: *APOBEC3A* = 0.0004; *GRHL3* <0.0001; *IVL* <0.0001; *ELF3* = 0.0072. Adjusted $P$ values for comparisons between NC#1 and *GRHL3* siRNA #2: *APOBEC3A* = 0.0012; *GRHL3* <0.0001; *IVL* <0.0001; *ELF3* = 0.0052. Cells were treated with 100 nM afatinib for 24 h prior to harvesting to induce differentiation. (E) Percentage of DDOST transcripts that were C > U edited at c558 in NIKS transfected with control (NC#1) or *GRHL3*-specific siRNAs as indicated ($n = 3$). Adjusted $P$ values: NC#1 vs *GRHL3* siRNA #1 = 0.0072; NC#1 vs *GRHL3* siRNA #2 = 0.0099. Gene expression (D) and DDOST editing (E) in *GRHL3* siRNA-transfected cells were compared with control siRNA-transfected cells using one-way ANOVA with Tukey's multiple comparisons test (error bars represent SEM; **$P < 0.01$, ***$P < 0.001$ and ****$P < 0.0001$). (F) ChIP-seq data for GHRL3, WDR5, H3K27Ac, H3K4Me1 and H3K4Me3 from keratinocytes and or monocytes and GeneHancer predicted regulatory regions (grey = enhancer, red = promoter) as indicated, spanning the *APOBEC3A* gene and a 33 kb region upstream of the TSS (see main text for references to the datasets). (G) Stacked barplot showing the proportion of each cell type in the single-cell atlas of healthy human airways from Deprez et al that express each selected gene. Source data are available online for this figure.

target sites to enable H3K4 methylation (Hopkin et al, 2012), and a WDR5 binding peak coincided with the GRHL3 peak at −33kb in differentiating NHEKs (Fig. 4F). A predicted promoter at −4kb relative to the TSS harbours NFκB and STAT2 binding sites previously implicated in *APOBEC3A* regulation (Oh et al, 2021; Isozaki et al, 2023), and WDR5 binding was also observed at this region in differentiating NHEKs (Fig. 4F; Appendix Fig. S7A). Analysis of ChIP-seq data from keratinocytes (Kouwenhoven et al, 2015; Rubin et al, 2017; Bernstein et al, 2010; Ernst et al, 2011) revealed a dominant peak for active histone marks (H3K27Ac, H3K4Me1 and H3K4Me3) at the −33kb enhancer where GRHL3 binding was detected (Fig. 4F). ChIP-seq data from monocytes (Nurminen et al, 2018; Bernstein et al, 2010; Novakovic et al, 2016; Dunham et al, 2012), which also express *APOBEC3A* (Peng et al, 2007; Chen et al, 2006; Koning et al, 2009) but not *GRHL3* (Kudryavtseva et al, 2003), revealed a different profile, with abundant H3K27 acetylation and H3K4 methylation marking the −4kb promoter and a second predicted enhancer at −15kb but not at the −33kb enhancer (Fig. 4F). Differential enhancer usage in keratinocytes and monocytes may explain the requirement for GHRL3 in the former but not the latter. GRHL3 and WDR5 binding to a region of the *ELF3* promoter containing three putative GRHL3 binding motifs was also evident (Appendix Fig. S7B), but no peaks were observed at the *IVL* promoter.

The relationship between *GRHL3* and *APOBEC3A* was further investigated in a dataset of 77,969 cells from healthy human airways (Deprez et al, 2020). Both *GRHL3* and *APOBEC3A* were predominantly (865/1545 (56%) of *GHRL3*-expressing cells and 230/623 (37%) of *APOBEC3A*-expressing cells) expressed in secretory epithelial cells of the nasal cavity, which also expressed the differentiation markers *KRT10, IVL* and *ELF3* (Fig. 4G). *APOBEC3B, MKI67* and *TOP2A* were predominantly expressed in cycling basal cells and suprabasal cells, again consistent with our observations from tonsil epithelium (Fig. EV1). As expected, *APOBEC3A* was also expressed in monocytes and macrophages (9.8 and 9.5% of *APOBEC3A*-expressing cells, respectively), while *GRHL3* was not (Fig. 4G). Together, these data from tissues and cultured cells identify GRHL3 as a key transcription factor that acts to upregulate *APOBEC3A* expression during keratinocyte differentiation.

Although no single transcription factor activity displayed a consistently strong correlation with *APOBEC3B* expression (Dataset EV2), possibly due to the lower number of *APOBEC3B*-expressing cells in several of the datasets, correlations with E2F family transcription factors are consistent with previous reports linking repressive E2F-containing complexes to *APOBEC3B* regulation (Periyasamy et al, 2017; Roelofs et al, 2020), and with its expression in proliferating cells.

## Evidence for regulation of *APOBEC3A* by GRHL3 in SCC tissues and cell lines

Having determined that *APOBEC3A* is expressed during the terminal differentiation of non-cancerous epithelial cells in the tonsil and oesophagus, and that this expression pattern could be recapitulated in immortalised but non-transformed epidermal keratinocytes in culture, we next investigated *APOBEC3A* and *APOBEC3B* expression patterns in scRNA-seq data from tumour samples. After inferring copy number variation (CNV) profiles for each cell by averaging gene expression levels over large genomic regions (Patel et al, 2014) (see methods for details), we identified 695 epithelial cells from our tumour samples with inferred CNV profiles that closely resembled those of cells from our matched normal tonsil cells (Appendix Fig. S8). These were classed as non-malignant and further analyses were based on the 18,619 malignant epithelial cells remaining. Pathway analysis of the top 100 genes co-expressed with either *APOBEC3A* or *APOBEC3B* in the 18,619 malignant tumour cells from the 10 Southampton HNSCCs (seven of which were patient-matched with the healthy tonsil samples analysed in Fig. 2 (Tables EV1 and EV2)) and in the additional published scRNA-seq datasets from HNSCC (Puram et al, 2023; Kürten et al, 2021) and ESCC (Zhang et al, 2021) revealed similar results to those obtained when performing the analysis on data from healthy tonsil; *APOBEC3A* was again co-expressed with genes in pathways related to keratinocyte differentiation, while *APOBEC3B* was co-expressed with genes in pathways linked to cell division (Appendix Fig. S9A–D; Datasets EV3–6).

Although it was not possible to visualise the four phenotypes (basal, proliferating, differentiating and terminally differentiated) on UMAPs due to the cells from individual tumours clustering by

patient rather than by phenotype (Appendix Fig. S9E), we again observed *APOBEC3A* co-expression with markers of differentiation and components of the RIPK4 pathway and *APOBEC3B* co-expression with markers of proliferation (Appendix Fig. S10A). When analysing each of the ten tumour samples in the Southampton HNSCC dataset individually, the same trends were observed in almost all cases (Appendix Fig. S10B–K). SCENIC analysis of the four SCC datasets implicated GRHL3 as a potential regulator of *APOBEC3A* in squamous cell carcinoma as well as in healthy epithelia, with strong correlations between GRHL3 activity and *APOBEC3A* expression evident across all studies (Fig. 5A; Dataset EV2). Having observed strong correlations between *APOBE3CA* expression and *GRHL3* expression and predicted activity in single-cell data from HNSCC and ESCC, we next sought to determine whether GRHL3 may be involved in regulating *APOBEC3A* expression cancer arising in other tissues. Since too few tumour cells in the scRNA-seq datasets from lung, breast and urothelial cancer that we analysed (Table EV1) expressed *APOBEC3A* to permit co-expression analysis, we examined *APOBEC3A* expression (Fig. EV2A), *GRHL3* expression (Fig. EV2B) and GRHL3 activity (Fig. EV2C) in bulk RNA-seq data from The Cancer Genome Atlas (TCGA). Squamous cell carcinomas (notably of the head and neck (HNSC), cervix (CESC-SQ), oesophagus (ESCA-SQ), lung (LUSC) and bladder (BLCA) were among the cancers with highest *APOBEC3A* expression (Fig. EV2A), and these also displayed the highest GRHL3 expression (Fig. EV2B) and activity scores (as inferred from the expression of genes in the GRHL3 regulon identified in our earlier SCENIC analysis, Fig. EV2C). As previously reported (Green et al, 2017), *APOBEC3A* was also highly expressed in acute myeloid leukaemia (LAML), but *GRHL3* was not, which is consistent with our earlier conclusion that *APOBEC3A* is differentially regulated in keratinocytes and myeloid cells. We also observed strong correlations between *APOBEC3A* and *GRHL3* expression (Fig. EV2D), and GRHL3 activity scores (Fig. EV2E) in bulk RNA-seq data from SCCs, and from adenocarcinomas of the cervix (CESC-AD) and oesophagus (ESCA-AD). Other cancer types in which *APOBEC3A* has been implicated in somatic mutagenesis, such as breast cancer, lung adenocarcinoma and bladder cancer, displayed weaker but nonetheless significant correlations between *APOBEC3A* expression and *GRHL3* expression and/or transcriptional activity, suggesting GRHL3 may play a role in promoting *APOBEC3A* expression in these tumours. It is important to note the limitations of performing this analysis in bulk RNA-seq data, in which *APOBEC3A* expression in cell types (e.g. tumour-resident macrophages or infiltrating neutrophils) that do not express GRHL3 will influence these correlations. Given the strong evidence from both single cell and bulk RNA-seq data that implicates GRHL3 in *APOBEC3A* regulation in HNSCC, we next analysed RNA-seq data from the 34 HNSCC cell lines in the Cancer Cell Line Encyclopaedia (CCLE) (Ghandi et al, 2019), observing correlations between *APOBEC3A*, *GRHL3* and differentiation-related genes, and inverse correlations between *APOBEC3A* and proliferation markers, *MKI67*, *MCM7* and *TOP2A* (Fig. EV3A). Among those cell lines profiled by the CCLE, *APOBEC3A* and *GRHL3* mRNA levels were highest in BICR6 and BICR22; lines derived from an SCC of the hypopharynx and from a lymph node metastasis from a tongue SCC respectively (Edington et al, 1995) (Fig. EV3B). *APOBEC3A* and *GRHL3* mRNA levels were higher in sub-confluent cultures of both BICR6 and

BICR22 than in NIKS harvested under the same conditions (Appendix Fig. S11), and we observed a significant reduction in *APOBEC3A* expression upon *GRHL3* knockdown in both cell lines (Fig. 5B).

To gain further insight into the heterogeneity of *APOBEC3A* and *APOBEC3B* expression in HNSCC, we next analysed spatial transcriptomics data obtained from tissue sections representing the same cases as those from which our scRNA-seq were derived. Consistent with what we observed in the scRNA-seq analysis, *APOBEC3A* was expressed in regions that displayed high predicted GRHL3 activity (the GRHL3 target genes that comprise the GRHL3 module are listed in Table EV5) and expression of additional genes related to keratinocyte differentiation, while *APOBEC3B* was expressed in regions marked by high expression of proliferation markers (Fig. 5C). Pathway analysis of genes co-expressed with *APOBEC3A* or *APOBEC3B* yielded similar results to those obtained from the scRNA-seq data but in addition to pathways associated with keratinocyte differentiation, the wound healing response (another process in which GRHL3 plays a critical role) was also overrepresented among those genes co-expressed with *APOBEC3A* (Table EV6; Appendix Fig. S12). Since the Visium platform typically provides resolution of approximately 10 cells / spot depending on cell size and cellularity, we performed spot deconvolution, observing that the *APOBEC3A* reads from each spot were largely derived from epithelial (tumour) cells with expression also evident in monocytes and neutrophils, consistent with previous reports (Suspène et al, 2011; Aynaud et al, 2012; Chen et al, 2006). *APOBEC3B* reads were largely derived from the tumour cells (Appendix Fig. S13). A representative example tumour section (case HN485), displaying regions of *APOBEC3A* expression with high GRHL3 activity ('GRHL3 module', composed of SCENIC-predicted target genes, including *ELF3*) is shown in Fig. 5d. In the same section, distinct *MKI67*-positive regions of the tumour show peak expression of *APOBEC3B*. Strong *CDKN2A* expression (the gene encoding p16$^{INK4A}$, a biomarker for HPV-positive HNSCC) is evident throughout most of the tumour cells. GRHL3 activity and *APOBEC3A* expression were frequently highest near the tumour surface.

Analysis of APOBEC3 protein expression in tissue samples has been hampered by a lack of suitable antibodies for detection by immunohistochemistry, but we (MAC and RSH) recently developed a monoclonal antibody that specifically detects APOBEC3A in formalin-fixed, paraffin-embedded tissues (Naumann et al, 2023). Having confirmed specificity by staining of paraffin-embedded blocks generated from PMA-treated wild-type control and *APOBEC3A*-knockout (KO) NIKS (Appendix Fig. S14), we conducted APOBEC3A immunohistochemistry on a tissue microarray (TMA) representing 20 HNSCC cases (10 HPV+ve and 10 HPV−ve). As predicted from our scRNA-seq and spatial transcriptomics data, some tumours were devoid of APOBEC3A, while others displayed abundant staining in more differentiated tumour cells, including in those cells surrounding keratin pearls – a distinguishing feature of well-differentiated SCC (Fig. EV4(I), left panel). Staining the same TMA with an antibody that binds to APOBEC3A, APOBEC3B and APOBEC3G (Brown et al, 2019) revealed characteristic nuclear APOBEC3B expression in tumour cells (Fig. EV4, right panels). As expected, this was particularly evident in HPV-positive cases (Fig. EV4 (II, III), right panels), in which APOBEC3B is upregulated by the viral E6 and E7 proteins (Vieira et al, 2014; Warren et al, 2015; Periyasamy et al, 2017; Mori

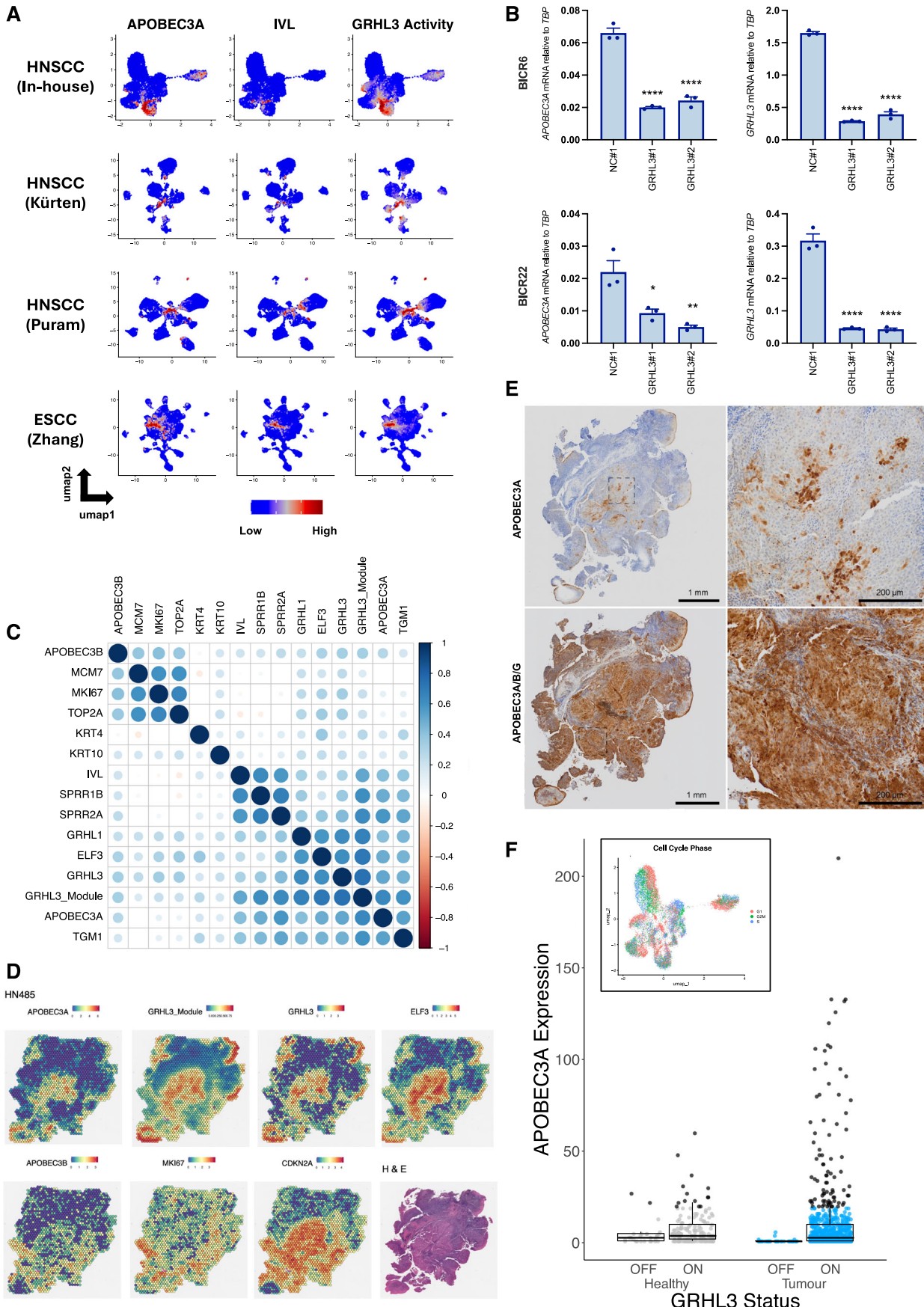

**Figure 5. GRHL3 regulates *APOBEC3A* expression in squamous cell carcinoma.**

(A) UMAPs heatmap showing gene expression of *APOBEC3A* and *IVL* and predicted activity of GRHL3 in scRNA-seq data from four independent tumour cohorts (three HNSCC and one ESCC). (B) qRT-PCR-based gene expression measurements of *APOBEC3A* and *GRHL3* in sub-confluent BICR6 (top row) and BICR22 (bottom row) HNSCC cells transfected with control (NC#1) or *GRHL3*-specific siRNAs as indicated. Adjusted *P* values: NC#1 vs *GRHL3* siRNA #1 (*APOBEC3A*: BICR6 <0.0001; BICR22 = 0.0145; *GRHL3*: BICR6 <0.0001; BICR22 <0.0001); NC#1 vs *GRHL3* siRNA #2 (*APOBEC3A*: BICR6 <0.0001; BICR22 = 0.0035; *GRHL3*: BICR6 <0.0001; BICR22 <0.0001). Gene expression in *GRHL3* siRNA-transfected cells was compared with control siRNA-transfected cells using one-way ANOVA with Tukey's multiple comparisons test ($n = 3$; error bars represent SEM; ****$P < 0.0001$; **$P < 0.01$ and *$P < 0.05$). (C) matrix showing the relationship between expression of the indicated genes in spatial transcriptomics data from the Southampton HNSCC cohort obtained using the Visium platform (10X Genomics). (D) Images displaying expression levels (Visium spot intensities) of selected genes in HN485, an HPV+ve HNSCC case from the Southampton cohort. (E) Immunohistochemistry with an antibody specific for APOBEC3A (left) and with an antibody that cross-reacts with APOBEC3A, APOBEC3B and APOBEC3G (right) in sections from the same tissue block from HN485 used for the Visium profiling displayed in part D. (F) Boxplot showing expression of *APOBEC3A* in those cells predicted to be in S-phase in normal tonsil and HNSCC, stratified by binary GRHL3 activity score (on/off). Cells shown in black are outliers relative to the distribution of expression in the cells from healthy tonsils. Inset: UMAP showing the predicted cell cycle phase for each cell in the Southampton HNSCC scRNA-seq dataset. The top, middle and bottom lines of the boxplot represent the upper quartile (Q3), median, and lower quartile (Q1), respectively. The maximum value represented by the top whisker represents the highest observed data point within $Q3 + (1.5 \times (Q3 - Q1))$, and the minimum value represented by the bottom whisker represents the lowest situated point within $Q1 - (1.5 \times (Q3 - Q1))$. Healthy GRHL3 'OFF', $n = 19$; Healthy GRHL3 'ON', $n = 173$; Tumour GRHL3 'OFF', $n = 98$; Tumour GRHL3 'ON', $n = 620$. Source data are available online for this figure.

et al, 2017). APOBEC3G is known to be expressed in the cytoplasm of T-lymphocytes and was evident in resident lymphocytes (e.g. Appendix Fig. S12b(I) arrowheads). The pan-cellular staining of keratinising cells with the APOBEC3A/B/G antibody is consistent with the APOBEC3A-specific staining (compare Fig. EV4(I) boxed areas and EV4(III) insets between left and right panels). We also stained three cases (HN485, Fig. 5E), HN482 and HN494 (Appendix Fig. S15), for which we had also generated spatial transcriptomics data, observing good concordance between the patterns of mRNA and protein positivity for both *APOBEC3A* and *APOBEC3B* (compare Fig. 5D,E; Appendix Fig. S13A,B). Importantly, in addition to providing further validation of the specificity of our antibodies, these data confirm that our conclusions relating to *APOBEC3A* and *APOBEC3B* expression drawn from mRNA data (scRNA-seq, spatial transcriptomics) are valid at the protein level.

Finally, while *APOBEC3A* expression was largely confined to *IVL*[+ve]/*MKI67*[−ve] (non-cycling) tumour cells, the correlation between *APOBEC3A* and *IVL* expression was weaker in the SCC datasets than in the normal tonsil epithelial cells, and in the UMAPs from the Southampton HNSCC dataset, *APOBEC3A* expression was apparent in *IVL*[−ve] cells, which also displayed high predicted GRHL3 activity (Fig. 5A). This was most obvious in two tumours (HN489 and HN492; compare Fig. 5A and Fig. S9E), suggesting that under certain conditions, activation of GRHL3 may induce *APOBEC3A* in cycling tumour cells. Given the considerable evidence linking APOBEC3A-mediated mutagenesis to deamination of the lagging strand during DNA replication (Green et al, 2016; Haradhvala et al, 2016; Hoopes et al, 2016; Morganella et al, 2016; Seplyarskiy et al, 2016; Stewart et al, 2020), we used gene expression data to assign cells from our normal tonsil and HNSCC datasets to G0/G1, S, or G2/M phase of the cell cycle (Fig. 5F inset) and compared *APOBEC3A* expression and GRHL3 activity in those cells predicted to be in S-phase. While, as expected, the majority of S-phase cells did not express *APOBEC3A*, considering all S-phase cells in which *APOBEC3A* expression was detectable (more than zero reads), we observed a small minority that expressed considerably more *APOBEC3A* mRNA than was seen in S-phase cells from normal tonsil epithelium (Fig. 5F; black dots on the boxplot represent cells that are statistical outliers with respect to the level of *APOBEC3A* found in normal tonsil). The S-phase tumour cells with high *APOBEC3A* expression were all designated a binary GRHL3 activity score of 'on', suggesting that GRHL3 can drive

*APOBEC3A* expression in tumour cells undergoing DNA replication, potentially causing APOBEC3A-mediated mutagenesis. The fact that we only observed high *APOBEC3A* expression in a small minority of S-phase cells in our HNSCC samples is consistent with the proposed episodic nature of APOBEC-mediated mutagenesis, in which the chances of observing a mutagenic burst in the snapshot provided by a tumour biopsy are low (Petljak et al, 2019, 2022a, 2022b; Smith and Fenton, 2019).

## Discussion

Our analysis of *APOBEC3A* and *APOBEC3B* gene expression in healthy and cancerous squamous epithelia provides new insight into how these genes are regulated and raises several questions that warrant further investigation. The low expression of *APOBEC3B* in normal epithelium and increased levels in tumours that we observed in scRNA-seq datasets is consistent with previous analyses of bulk tissue samples and breast cancer cell lines (Burns et al, 2013a), in which repressive E2F/RB complexes have been shown to silence expression in quiescent cells (Periyasamy et al, 2017; Roelofs et al, 2023, 2020). Loss of p53-mediated repression of *APOBEC3B* transcription, resulting either from *TP53* mutation (observed at high frequency in HPV-negative HNSCC and ESCC) or from HPV E6/E7 activity in HPV-positive HNSCC (Vieira et al, 2014; Periyasamy et al, 2017) is also likely an important driver of *APOBEC3B* expression seen in many of the SCC samples we analysed.

The high expression of *APOBEC3A* in tonsil and oesophageal epithelium could indicate a role in defence against one or more viruses with tropism for the upper-aerodigestive tract. Wild-type adeno-associated virus (AAV, a target for APOBEC3A (Chen et al, 2006)) infects keratinocytes via binding to heparan sulphate proteoglycans, and AAV genomic DNA has been isolated from tonsils (Schnepp et al, 2005). *APOBEC3A* and *APOBEC3B* have both been implicated in host responses to HPV infection (Wang et al, 2014; Vartanian et al, 2008; Vieira et al, 2014; Warren et al, 2015) and although our data suggest that neither are expressed in quiescent basal cells of the tonsil epithelium (the target cell for HPV infection), *APOBEC3B* is expressed in those cells undergoing division in the parabasal layer, while *APOBEC3A* is expressed in cells undergoing terminal differentiation; a pattern also evident in

an area of the normal stratified epithelium at the margin of an HPV-associated oropharyngeal SCC in which *APOBEC3A* and *APOBEC3B* were detected by RNA in situ hybridisation (Kono et al, 2020). Both genes are therefore expressed (at least in the absence of infection) under cellular conditions in which different stages of the HPV productive life cycle occur: genome maintenance following E6/E7-induced cell cycle entry in the basal/para-basal layer and genome amplification in terminally differentiating cells (Doorbar et al, 2012). Whether this pattern of APOBEC3 gene expression represents a host adaptation to papillomaviruses, or to other pathogens that infect the upper-aerodigestive tract remains to be determined. Alternatively, the expression of *APOBEC3B* in dividing keratinocytes and *APOBEC3A* during terminal differentiation may reflect hitherto unidentified physiological roles related to these processes. *APOBEC3B* is also expressed as breast cancer cell lines approach mitosis, and its knockdown slows proliferation, suggesting a role in cell cycle progression that might be linked to its function as a transcriptional co-activator for the oestrogen receptor (Periyasamy et al, 2015; Roelofs et al, 2023). Unlike in normal breast epithelium (Fig. 1) or in MCF10A, a cell line derived from the normal mammary epithelium (Roelofs et al, 2023), we observed sufficient *APOBEC3B* expression in our scRNA-seq data from normal tonsil epithelium and from NIKS (Fig. 3e) to observe a clear enrichment in G2/M-phase cells, suggesting APOBEC3B may play a role in normal keratinocytes entering cell division. *APOBEC3B* expression in G2/M-phase is not unique to epithelial cells either; it has also been documented in myeloma cells and in B-cells from healthy bone marrow (Hirabayashi et al, 2021). *APOBEC3A* induction during $Ca^{2+}$ induced keratinocyte differentiation has been linked to hypermutation of mitochondrial DNA (Wakae et al, 2018), although the functional significance of this remains unclear. More investigation of *APOBEC3A* and *APOBEC3B* function in epithelial cells is required, but it is maybe not surprising that by restricting *APOBEC3A* expression to post-mitotic keratinocytes and *APOBEC3B* expression to the G2/M phase of proliferating cells, mechanisms have evolved to confine these potentially dangerous deaminases to contexts in which DNA replication is not occurring.

The identification of GRHL3 as a transcription factor responsible for driving *APOBEC3A* expression in differentiating keratinocytes and in squamous cell carcinoma highlights the power of single-cell transcriptomics to uncover gene regulatory networks. In this case, using SCENIC (Aibar et al, 2017) we observed striking correlations between GRHL3 activity scores and *APOBEC3A* expression across multiple scRNA-seq datasets from healthy and cancerous epithelia and validated the prediction using RNA interference in cultured cells. GRHL3 is a key transcription factor in epidermal keratinocytes, required not only during differentiation but also in migration during developmental processes and in wound healing (Ting et al, 2005; Caddy et al, 2010; Hislop et al, 2008; Gordon et al, 2014; Yu et al, 2006, 2008). Given the extensive overlap between the molecular processes that are active during wound healing and cancer (MacCarthy-Morrogh and Martin, 2020), this latter function may be of particular relevance to driving *APOBEC3A* expression in tumour cells, including in those undergoing DNA replication and warrants further investigation.

While most studies have focused on its function in the epidermis, *GRHL3* and its murine orthologue, *Grhl3*, have been implicated as suppressors of squamous carcinogenesis in the mucosal epithelia of the oral cavity and oesophagus as well as in skin (Darido et al, 2011; Georgy et al, 2023). *GRHL3* loss-of-function mutations have not been reported in SCC, but it is located at a locus (1p36.11) that is frequently deleted in HNSCC, and it is also targeted by a micro-RNA (miR-21) that is over-expressed in HNSCC. Accordingly, *GRHL3* expression in SCCs was demonstrated to be significantly lower than in adjacent normal tissue (Georgy et al, 2015; Darido et al, 2011). Our scRNA-seq analysis of normal tonsil and HNSCC cases is in agreement with the above studies; we observed higher mean *APOBEC3A* expression in normal tonsil epithelial cells than in tumour cells from patient-matched HNSCC cases (Fig. 1). Similarly, while spatial transcriptomics and immunohistochemistry of sections from these HNSCCs revealed widespread *APOBEC3B* expression (particularly in HPV +ve cases, as expected), *APOBEC3A* was not expressed in all cases and in those tumours where expression was observed, it was restricted to areas of high GRHL3 activity (Fig. 5). Repression of *APOBEC3B* expression by the E2F4/RB-containing DREAM (dimerisation partner, RB-like, E2F and MuvB) complex has been reported (Periyasamy et al, 2017), which may explain its confinement to proliferating cells in healthy squamous epithelia. The absence of *APOBEC3A* in basal and terminally differentiated cells suggests it, too, is subject to transcriptional repression; another mode of regulation that, if disrupted, could enable bursts of mutagenic APOBEC activity in cancer cells.

These observations, together with our siRNA experiments in HNSCC cell lines, suggest that in SCC at least, *APOBEC3A* expression is confined to the minority of tumour cells in which *GRHL3* is expressed and active. If we consider that APOBEC activity is only likely to be mutagenic in cycling cells (or in cells that have re-entered the cell cycle without repairing deaminated cytosines), the pool of tumour cells at risk of APOBEC3A-mediated mutagenesis is likely limited to those rare cells highlighted in Fig. 5f, in which *APOBEC3A* expression coincides with DNA replication. It follows that if such *APOBEC3A*-expressing tumour cells were to acquire mutations that caused them to become more proliferative (as might be expected if the subclone were to expand to constitute a significant portion of the tumour), this would result in loss of *APOBEC3A* expression (and potentially an increase in *APOBEC3B* expression). This model could explain the somewhat puzzling observation that tumours with strong enrichment for mutational signatures (YpTp[C > T/C > G]pN) associated most strongly with APOBEC3A often express very low levels of *APOBEC3A*, while *APOBEC3B* expression is more closely correlated with enrichment for the APOBEC mutational signatures but appears to be responsible for generating a smaller fraction of these mutations (Chan et al, 2015; Jalili et al, 2020; Petljak et al, 2022a; Carpenter et al, 2023).

Finally, we note that activation of GRHL3 in keratinocytes plays a key role in resolving psoriatic lesions by suppressing inflammatory mediators and driving epidermal repair (Shi et al, 2021; Gordon et al, 2014). Our finding that GRHL3 regulates *APOBEC3A*, which was originally discovered as a protein (Phorbolin-1) that is highly upregulated in psoriatic lesions (Rasmussen and Celis, 1993; Madsen et al, 1999), finally provides a potential mechanistic explanation for this early observation that predated mapping of the *APOBEC3A* gene (Jarmuz et al, 2002) by almost a decade.

# Methods

## Reagents and tools table

| Reagent/resource | Reference or source | Identifier or catalogue number |
|---|---|---|
| **Experimental models** | | |
| NIKS | Allen-Hoffman et al J Invest Dermatol 114(3):444—55 (2000) | Cellosaurus Accession CVCL_A1LW |
| 3T3-J2 | Rheinwald J.G., Green H. Cell 6:331–343 (1975) | Cellosaurus Accession CVCL_W667 |
| Normal primary human keratinocytes from pooled adult donors (NHEK) | Sigma-Aldrich | C-12006 |
| **Recombinant DNA** | | |
| pCR Blunt II-TOPO | Thermo Fisher | 450245 |
| pX335-U6-Chimeric_BB-CBh-hSpCas9n(D10A) | Addgene (Cong et al Science 15;339(6121):819-23 (2013)) | 42335 |
| pEGFP-N1-flpo | This study | |
| *APOBEC3A* targeting plasmid (pBSA3A-del) | This study | |
| pRH3097-A3A (for qRT-PCR standard) | Reuben Harris / Refsland et al Nucl Acids Res 38(13):4274-84 (2010) | |
| pCMV4-APOBEC3B (for qRT-PCR standard) | Mike Malim (Kings College London, UK) | |
| **Antibodies** | | |
| APOBEC3A (rabbit monoclonal UMN-13) | In-house (Reuben Harris) / Naumann et al Int. J. Mol. Sci. 24(11), 9305 (2023) | |
| APOBEC3A/B/G (rabbit monoclonal 5210-87-13) | Cell Signaling Technology / Brown et al Antibodies 10;8(3):47 (2019)) | 81001 |
| **Oligonucleotides and other sequence-based reagents** | | |
| Primers and probes used for digital PCR, qRT-PCR and/or plasmid construction to generate standard curves. | *APOBEC3A, APOBEC3B* and *TBP* qRT-PCR primers: Refsland et al Nucl Acids Res 38(13):4274-84 (2010) *DDOST* primers and probes: Oh and Buisson STAR Protocols 3(1)101148 (2022) All other primers, this study All primers and probes were purchased from Integrated DNA Technologies. | Table 1 |
| SilencerSelect Non-targeting siRNA (NC#1) | Thermo Fisher | 4390843 |
| SilencerSelect GRHL3 siRNA #1 | Thermo Fisher | s33753 |
| SilencerSelect GRHL3 siRNA #2 | Thermo Fisher | s33754 |
| **Chemicals, Enzymes and other reagents** | | |
| **Inhibitors:** | | |
| Afatinib | Fisher Scientific | 16463748 |
| Everolimus | Fluka Analytics | 57-55-6 |
| GÖ6983 | Sigma-Aldrich | G1918 |
| MK2206 | Selleck Chemicals | S1078 |

| Reagent/resource | Reference or source | Identifier or catalogue number |
|---|---|---|
| Phorbol 12-myristate 13-acetate (PMA) | Stem Cell Technologies | 74042 |
| Pictilosib (GDC0941) | Selleck Chemicals | S1065 |
| Ravoxertinib (GDC0994) | Selleck Chemicals | S7554 |
| Trametinib (GSK1120212) | Selleck Chemicals | NC0991754 |
| **Cell culture:** | | |
| Adenine | Sigma-Aldrich | A2786 |
| Cholera Toxin | Merck-Millipore | 227036 |
| Dulbecco's modified eagle medium (DMEM) with 4.5 g/L glucose, 3.7 g/L NaHCO$_3$, with L-glutamine, without sodium pyruvate | PAN Biotech | P04-03550 |
| Epidermal growth factor (EGF) from murine submaxillary gland | Sigma-Aldrich | E1257 |
| Foetal bovine serum | PAN Biotech | P30-3031 |
| Ham's F12 medium, with 1.176 g/L NaHCO$_3$ and L-glutamine | PAN Biotech | P04-14500 |
| Hydrocortisone | Sigma-Aldrich | H0888 |
| Insulin | Sigma-Aldrich | I6634 |
| Mitomycin C from *Streptomyces caespitosus* | Sigma-Aldrich | M4287 |
| Penicillin / Streptomycin (containing 10,000 U/ml Penicillin and 10 mg/ml Streptomycin) | PAN Biotech | P06-07100 |
| Trypsin 0.05% / ethylene-diamine tetra acetic acid (EDTA) 0.02% in PBS | PAN Biotech | P10-0235SP |
| Lipofectamine™ RNAiMax | Thermo Fisher | 13778150 |
| FuGENE® HD | Promega | E2311 |
| **Molecular biology** | | |
| Kapa HiFi HotStart ReadyMix | Roche | KK2601 |
| Bbs I | New England Biolabs | R0539S |
| NEBuilder HiFi DNA assembly cloning kit | New England Biolabs | EE5520 |
| Monarch Total RNA Miniprep Kit | New England Biolabs | T2010S |
| LunaScript® Reverse Transcriptase (RT) SuperMix | New England Biolabs | E3010S |
| PowerUp™ SYBR™ Green Master Mix for qPCR | Thermo Fisher | A25742 |
| Absolute Q™ DNA Digital PCR Master Mix (5X) | Thermo Fisher | A52490 |
| **Software** | | |
| GraphPad Prism 10 | GraphPad Software LLC | |
| FACSDiva | BD Biosciences | |
| R-Studio | Posit (open source) | |
| R | R-project.org (open source) | |
| Cell Ranger | 10X Genomics | |
| **Other** | | |

**Table 1. Primers used for qPCR, digital PCR, genotyping and/or plasmid construction.**

| Gene | Sequence |
| --- | --- |
| APOBEC3A (Forward) | 5'-GAGAAGGGACAAGCACATGG-3' |
| APOBEC3A (Reverse) | 5'-TGGATCCATCAAGTGTCTGG-3' |
| APOBEC3B (Forward) | 5'-GACCCTTTGGTCCTTCGAC-3' |
| APOBEC3B (Reverse) | 5'-GCACAGCCCCAGGAGAAG-3' |
| KRT10 (Forward) | 5'-CCTGCTTCAGATCGACAATGCC-3' |
| KRT10 (Reverse) | 5'-ATCTCCAGGTCAGCCTTGGTCA-3' |
| IVL (Forward) | 5'-GGTCCAAGACATTCAACCAGCC-3' |
| IVL (Reverse) | 5'-TCTGGACACTGCGGGTGGTTAT-3' |
| MKI67 (Forward) | 5'-GAAAGAGTGGCAACCTGCCTTC-3' |
| MKI67 (Reverse) | 5'-GCACCAAGTTTTACTACATCTGCC-3' |
| MCM7 (Forward) | 5'-GCCAAGTCTCAGCTCCTGTCAT-3' |
| MCM7 (Reverse) | 5'-CCTCTAAGGTCAGTTCTCCACTC-3' |
| ELF3 (Forward) | 5'-CATGACCTACGAGAAGCTGAGC-3' |
| ELF3 (Reverse) | 5'-GACTCTGGAGAACCTCTTCCTC-3' |
| GRHL3 (Forward) | 5'-ACTGTGGAGCACATTGAGGAGG-3' |
| GRHL3 (Reverse) | 5'-CTGTGCTCAGACAGTTTACGCC-3' |
| TBP (Forward) | 5'-TTGAGGAAGTTGCTGAGAAGAG-3' |
| TBP (Reverse) | 5'-CAGATAGCAGCACGGTATGAG-3' |
| TBP standard (Forward) | 5'-CACTCACAGACTCTCACAACTG-3' |
| TBP standard (Reverse) | 5'-GTCGTCTTCCTGAATCCCTTTAG-3' |
| DDOST (Forward) | 5'- ACTGAGAACCTGCTGAAG-3' |
| DDOST (Reverse) | 5'- AAGAGGATGGGATTTAGAGA-3' |
| DDOSTC558 Probe | 5'-(HEX)-CAACCATCGTTGGGAAATC-(Q)-3' |
| DDOSTT558 Probe | 5'-(FAM)-CCAACCATTGTTGGGAAATC-(Q)-3' |
| A3A N-term left sgRNA (sense) | 5'- CACCGCTTGCGACTTGCTCAAGGCG-3' |
| A3A N-term left sgRNA (antisense) | 5'-AAACCGCCTTGAGCAAGTCGCAAGC-3' |
| A3A N-term right sgRNA (sense) | 5'-CACCGCACAGACCAGGAACCGAGAA-3' |
| A3A N-term right sgRNA (antisense) | 5'-AAACTTCTCGGTTCCTGGTCTGTGC-3' |
| A3A deletion genotyping primer (Forward) | 5'-TGAGCTCACACCAGAACCAC-3' |
| A3A deletion genotyping primer (Reverse) | 5'-TAGAGCCCAGAGAAGGTCCC-3' |

## Ethics/patient samples

Patients undergoing biopsies of suspected primary Head and Neck cancers at University Hospitals Dorset (UHD) NHS Foundation Trust were consented to take part in a study; 'Head and Neck cancer: molecular, cellular and immunological mechanisms'. This study is the NIHR portfolio adopted (portfolio No. 8130) and has been approved by the National Research Ethics Service South Central committee (reference No. 09/H0501/90). Tumour samples from ten oropharyngeal patients (Table EV1), as well as normal tissue samples from the contralateral tonsil for seven of the patients (collected at the time of diagnostic biopsy), were selected for single-cell RNA sequencing. Informed consent was obtained from all subjects and the experiments conformed to the principles set out in the WMA Declaration of Helsinki and the Department of Health and Human Services Belmont Report. Sample number (17), including ten tumour samples and seven matched tumour-free contralateral tonsil controls, was dictated by sample availability and funding as opposed to statistical considerations.

## Single-cell suspension preparation

Upon receipt, tissue samples were washed once in Dulbecco's modified eagle medium (Sigma #D5671) containing 10% foetal calf serum, 1% Penicillin/streptomycin, 1% L-glutamine, 1% amphotericin, 1% sodium pyruvate and 12.5 mM HEPES. Samples were chopped into 1–2 mm size pieces prior to enzymatic digestion. The first stage of the enzymatic digestion was performed using Liberase™ (Sigma #5401020001) at 100 µg.mL$^{-1}$ and DNase-1 (Sigma #DN25) at 16 units.mL$^{-1}$ in cDMEM. The solution was sterile filtered using a 0.22 µm syringe filter, and the sample material was suspended in up to 5 mL of cDMEM/Liberase solution. Samples were then sealed and placed in a benchtop shaker/incubator at 37 °C and 150 rpm for 15 min and then removed. The tube was left to stand until the undigested material had settled to the bottom then the upper 4–4.5 mL was carefully transferred to a fresh tube, the Liberase fraction. For the second digest (Col+) cDMEM containing collagenase-P (Sigma #11213857001) at 3 units.ml$^{-1}$, liberase at 100 µg.mL$^{-1}$, dispase (Sigma #D4693) at 0.5 units.mL$^{-1}$, elastase (Sigma #E1250) at 400 µg.mL$^{-1}$, trypsin (Sigma #T4799) at a final concentration of 0.25%, and DNase-1 (16 units.mL$^{-1}$) was added to the remaining material through a 0.22 µm sterile syringe filter. The Col+ digest was returned to the incubator (37 °C/150 rpm) for up to a maximum of 45 min (or until digestion is complete), with trituration performed using a 5 mL graduated pipette every 15 min. After 45 min, the Col+ fraction was removed from the incubator and any remaining undigested pieces were allowed to settle at the bottom of the tube; the supernatant was then transferred to a fresh sterile tube. Any remaining tissue was set aside.

The post-digestion process was the same for both the Liberase and Col+ fractions. Complete DMEM, up to 10 mL, was added to each fraction, and both cell suspensions were pelleted at 350 rcf for 5 min. The supernatant was removed and RBC lysis buffer (BioLegend #420301) was used to remove erythrocytes for 10 min at 4 °C. The samples were then washed in PBS, suspended in residual volume and then held at 4 °C until the Col+ fraction was prepared. Cell pellets were suspended in PBS containing 2% BSA-Fraction V (Scientific Lab Supplies #10735108001) and passed through a pre-wetted 40 µm filter. Both samples were then counted, and viability was assessed by Trypan blue exclusion. A final visual check of sample quality was also performed to ensure there were no large clumps of cells nor debris from the digestion. Finally, the two fractions were used to make a 100 µL suspension of 100,000 cells, of which 10,000 were from the liberase fraction, and 90,000 from the Col+ fraction, and 2% BSA in PBS was used as the diluent. This cell suspension was then run immediately on a Chromium Controller (10X Genomics).

## Fluorescence-activated cell sorting

Flow cytometry to determine the proportions of cell types in the disaggregated samples was carried out using a FACSCanto II (BD

Biosciences). Cell viability was assessed using Zombie Violet™ (Biolegend #423114). The following antibodies were purchased from Biolegend: EpCAM (#369806), CD90 (#328114), CD45 (#368508), CD31 (#303118) and CD3 (#300426). A minimum of 20,000 events were acquired for each case. Gating was applied to exclude debris, dead cells, doublets and the immune compartment (CD45+ and/or CD3+) before enumerating the numbers of endothelial, epithelial, and CD90-positive fibroblasts in the sample.

## Single-cell RNA sequencing

Five thousand single cells from each sample were captured on a Chromium Controller™ (10X Genomics) system using Illumina single cell 3' gene expression and library preparation kits (V3.1 #1000269). Sample capture, sample indexing, and library preparation were carried out strictly according to the manufacturer's instructions. Size distribution, quality control, and quantification of the libraries was assessed using High Sensitivity DNA chips (Agilent Technologies #5067-4626) and KAPA library quantification qPCR kit (Roche #07960140001). Prepared libraries were pooled and sent to Oxford Genomics (UK) for 150-base pair, paired-end sequencing on a Novaseq6000™.

## Sequence alignment and annotation

Cell Ranger (10x Genomics) pipelines (mkfastq, count) were used to align reads, filter, count barcodes and UMIs (unique molecular identifiers) and generate feature-barcode matrices. FASTQ files were aligned to the Human reference genome (GRCh38–2020-A), which had the HPV genome concatenated to both FASTA and GTF reference files. HPV reference sequences were downloaded from PaVE: The Papillomavirus Episteme (https://pave.niaid.nih.gov). The HPV16 reference sequence (NC_001526) was used in the first instance and in cases requiring further identification of the HPV subtype references including HPV-33 (OQ_672679) and HPV-18 (NC_001357) were also created. In all cases, the individual HPV ORFs were identified in the FASTA and GTF files to allow identification during alignment.

## Pre-processing of scRNA-seq data

For each sample, raw gene expression matrices were integrated into one dataset using the Seurat package (v4.0.1). The resulting feature-barcode matrix from the cell ranger pipeline was transformed into a Seurat object with patient metadata. Cells with less than 200 expressed genes were removed. Genes expressed in less than 3 cells were also filtered out. Further low-quality cells were removed based on mitochondrial gene percentage with the threshold calculated as the median + 3*median absolute deviation. Cells above the threshold were removed, ensuring high-quality cells remained.

## Normalisation and integration

After quality control steps, the data was normalised to adjust for differences in sequencing depth between samples. sctransform was chosen to normalise, and variance stabilise the count data109. Reciprocal PCA ('RPCA') Seurat integration workflows were utilised for integration. The Seurat object was first split by the patient into a list of ten smaller objects, in which each dataset was normalised by sctransfrom individually. 3000 features were selected via 'SelectIntegrationFeatures' function. 'PrepSCTIntegration' was run prior to anchor identification to ensure sctransfrom residuals from the 3000 features identified (by SelectIntegrationFeatures) were present. Anchors, used to integrate objects, were found between datasets using FindIntegrationAnchors, with the normalisation.method set to 'SCT' and reduction set to 'rpca' (all other parameters were default). Finally, IntegrateData was run, again specifying 'SCT' as normalisation.method. This integration pipeline was run using IRIDIS High Performance Computing Facility (University of Southampton).

## Dimensionality reduction, visualisation and clustering

Principal component analysis (PCA) was used to reduce the dimensionality of the datasets. Principal components were assessed by JackStraw and elbow plots to select an appropriate number of dimensions to be used downstream. Dimensions 1:30 were selected in the following steps. Clustering was performed in Seurat, which constructs a k-nearest neighbours graph and refines this using the shared local neighbourhood overlap between cells ('FindNeighbours'; 'FindClusters'). RunUMAP command was used to visualise the data in a UMAP (Uniform, Manifold, Approximation and Projection) plot.

## Identification of marker genes and cell type identification

After clustering and UMAP projection, broad cell populations were identified based on the expression of known marker genes e.g. PTPRC/CD45+ immune cells, LUM+ Fibroblasts, RGS5+ Mural cells and PECAM1/CD31+ endothelial cells. Epithelial cell clusters were identified based on the expression of EPCAM, SFN and cytokeratin genes (e.g. KRT14, KRT17, KRT6A, KRT5 and KRT19) —with the absence of expression of other cell-type markers. Epithelial cell clusters were then subset into a separate object, with new variable features found by re-running sctransform, PCA and clustering, whereby any clusters suggestive of doublets were removed based on the expression of non-epithelial markers (identified using FindAllMarkers) and examining UMI/feature number. The remaining epithelial cells were then used for further analysis.

## Unsupervised clustering of epithelial cells

A total of 22,595 epithelial cells were subset into tumour (19,314 cells) and normal (3281 cells) Seurat objects. The normal epithelial cells underwent further quality control, 658 cells were removed as suspected doublets due to high expression of immune cell and fibroblast related genes. The remaining 2649 cells were clustered using the first seven principal components with a resolution of 0.2, and a k parameter of 30. Cell subtypes were identified using known gene markers for epithelial and keratinocyte cell states, and a priori knowledge. The density of APOBEC3A and APOBEC3B expression was visualised on UMAPs using the Nebulosa R package (v1.9.0). R package inferCNV (v1.3.3) was used to identify malignant and non-malignant cells. The epithelial cells from our matched normal tonsil samples (grouped by differentiation status: basal; proliferating; differentiating and terminally differentiated) were used as reference cells, and the infercnv::run function was used with a cutoff of 0.1

and HMM = FALSE, and clustered by the patient (Appendix Fig. S8A). An inferCNV score (number of genes with inferred CNV/total number of genes) was calculated for each cell. The higher the score, the more likely a cell was malignant. The distribution of inferCNV scores was clearly bimodal, enabling us to set a threshold for malignant cells at the lowest point between the two distributions (0.12, Appendix Fig. S8B). This resulted in 695 non-malignant cells that were removed from our analyses of gene expression in tumour cells and 18,619 malignant cells that formed our tumour cell dataset. The tumour cells were clustered using the first 15 principal components with a resolution of 0.2 and k parameter of 60.

## Gene co-expression

The COTAN R package (v2.0.1) was used to investigate the co-expression of gene pairs for scRNA-seq datasets. For both APOBEC3A and APOBEC3B, the top 100 genes with a positive correlation index were identified and used in pathway analysis using the enrichR package (v3.2) and GO biological processes gene sets. Relevant epithelial differentiation and proliferation markers were chosen, and APOBEC3A/APOBEC3B co-expression values with selected genes were plotted in heatmaps using R package pheatmap (v.1.0.12).

## SCENIC analysis

Transcription factor (TF) analysis of scRNA-seq data was performed using pySCENIC (v0.12.1) (Kumar et al, 2021) and motif collection version mc9nr. TF activity AUC score for each cell was overlaid on UMAPs for visualisation, and the score was correlated with APOBEC3A and APOBEC3B expression using Spearman correlation and corrected for multiple tests using Benjamini–Hochberg. The binary activity (on/off) of each TF was determined based on a threshold generated by pySCENIC, and each cell was classified as on (1) or off (0). The APOBEC3A levels in 'GRHL3 on' and 'GRHL3 off' cells were compared using the Wilcoxon rank-sum test. Four groups of APOBEC3A/GRHL3 expression were considered: $APOBEC3A^-$/ $GRHL3^-$, $APOBEC3A^+$/$GRHL3^-$, $APOBEC3A^-$/$GRHL3^+$, and $APOBEC3A^+$/$GRHL3^+$. The number of cells in each group were counted and a comparison between the number of 'GRHL3 on' ($GRHL3^+$) cells that were also $APOBEC3A^+$ were compared with the number of 'GRHL3 off' ($GRHL3^-$) cells that were $APOBEC3A^+$ using Fisher's exact test. The FindAllMarkers function in the Seurat R package was used to perform the differential transcription factor activity analysis, using a threshold fold change of 1.1 and Benjamini–Hochberg adjusted P of 0.05.

## GHRL3 binding motif analysis

Homer (v4.11) function findMotifsGenome with mismatches threshold set to 2 was used to identify the frequency of GRHL3 binding motifs in differentiating NHEK ChIP-seq for GRHL3 peaks. The three binding motifs searched for were based on previous literature: AACCGGTT (Boglev et al, 2011), AACCTGTT and AACAGGTT (Yu et al, 2006). The percentage of times each base was located at each position in the binding motif was calculated and visualised using R package motifStack (v1.44.1) and dependencies. GRHL3 and WDR5 ChIP-seq for differentiating and proliferating keratinocytes ChIP-seq data (bedfiles), and monocyte and keratinocyte H3K27Ac, H3K4Me1 and H3K4Me3 ChIP-seq data (BigWig files) was obtained from CistromeDB and visualised in the UCSC genome browser.

## Airways single-cell atlas analysis

Publicly available data for healthy human airways (HG38 version) was obtained from https://doi.org/10.1164/rccm.201911-2199OC. Only cells expressing APOBEC3A (623 cells), APOBEC3B (1008), GRHL3 (1545), ELF3 (45,597), IVL (113), KRT10 (52,987), MKI67 (730) and TOP2A (1060) were considered for individual analysis. Proportions of each cell type that a gene was expressed in was calculated and visualised in stacked bar plots.

## Spatial transcriptomic analysis

All steps leading up to sequencing (from the bench side) were performed per manufacturer recommendations on 6.5 mm capture areas using the Visium V2 cytassist workflow. All the samples were run through the Spaceranger pipeline (v2.0.0) as per 10X Genomics/Visium guidelines.

Count matrices were loaded into Seurat. Samples were normalised using the SCTransform (Hafemeister and Satija, 2019) function (using the variance-stabilising transformation). To identify spot clusters across patients, samples were integrated using the Seurat v3 CCA anchor finding method (FindIntegrationAnchors and IntegrateData). The 3000 variable features selected for integration were then used for principal component analysis (PCA), followed by FindNeighbors and FindClusters for (shared) nearest-neighbour graph construction and cluster determination, respectively. Uniform manifold approximation and projection (UMAP) algorithm (1:20 dimensions) was used to visualise the batch-corrected integrated dataset. The resulting clusters were inspected, with poor quality clusters removed. The GRHL3 module score for each spot was calculated using AddModuleScore with 127 genes identified by SCENIC as potential target genes for GRHL3 binding in normal epithelium and HNSCC. Spatial feature expression plots were generated with the SpatialFeaturePlot function in Seurat.

Robust Cell Type Decomposition (RCTD) (Cable et al, 2021) with spacexr 2.2.1111 in R was used to deconvolve Visium spots into cell types using the annotated scRNA-Seq HNSCC reference dataset. RCTD was run with default parameters and doublet mode set to 'full' on each individual patient sample, with resulting deconvoluted normalised weights for each cell type obtained.

## APOBEC3A expression in cancer cell lines

Expression data for 34 head and neck cancer cell lines from the Cancer Cell Line Encyclopaedia (CCLE) (Barretina et al, 2012) was obtained from the resource CellminerCDB (Luna et al, 2021) for differentiation and proliferation marker genes, APOBEC3A, and genes in the RIPK4 pathway. Spearman correlation coefficients were calculated pairwise for all genes.

## TCGA RNA-seq analysis

Data were collected from the UCSC Xena functional genomics browser [https://xenabrowser.net]. Batch-corrected gene expression

data (RNAseq, log2(normalised value + 1)) for 11,060 patients (version 2016-12-29), and clinical data for 12,591 patients (version 2018-09-13) were downloaded for the TCGA pancancer (PAN-CAN) cohort (31 cancer types). Cervical (CESC) and oesophageal (ESCA) cancer cohorts were separated into squamous (CESC-SQ and ESCA-SQ) and adenocarcinoma (CESC-AD and ESCA-AD). *APOBEC3A* and *GRHL3* expression was visualised in boxplots made with R package ggplot2(v3.5.1) across all cancer types, sorting from the highest median expression to the lowest. A GRHL3 signature was calculated for each sample by summing the means of the 127 genes predicted as being regulated by *GRHL3* in the previously described SCENIC analysis. This score was visualised in boxplots for each cancer type.

## Tissue microarrays

Tissue microarrays (TMAs) were constructed from paraffin-embedded HNSCC and normal oral mucosa (10 HPV+ve HNSCC, 10 HPV-ve HNSCC, 10 fibroepithelial polyps) using triplicate, randomly selected, 1-mm tumour cores (Aphelys Minicore 2, Mitogen, Harpenden, UK). Automated immunostaining (DAKO/Agilent Autostainer) was performed in a CPA-accredited clinical cellular pathology department.

## Immunohistochemistry

Staining of tissue microarrays and full-face sections was performed on a Dako link automated staining machine according to the manufacturer's instructions. The following antibodies were used: rabbit monoclonal anti-human APOBEC3A/B/G (Cell Signaling Technology, Cat#81001; 1:100 dilution with Dako FLEX TRS high pH retrieval); rabbit monoclonal anti-human APOBEC3A (UMN-13); 1:1000 dilution with Dako FLEX TRS high pH retrieval). Slides were scanned on an Olympus VS110 Digital Slide Scanner using a 20x magnification air objective. Staining and analysis of staining was performed blinded to the results of spatial transcriptomics profiling of adjacent sections.

## Cell culture

Low-passage Normal immortalised keratinocytes (NIKS) and Normal primary human keratinocytes from pooled adult donors (Sigma C-12006) were cultured in FC medium (3:1 Ham's F12:DMEM, 5% foetal bovine serum (FBS), 10 ng/ml murine submaxillary gland EGF, 24 μg/ml adenine, 5 μg/ml insulin, 8.3 ng/ml cholera toxin, 0.4 μg/ml hydrocortisone, 1% penicillin/streptomycin) on a feeder layer of mitomycin C-treated mouse embryonic fibroblasts (3T3-J2). BICR6 and BICR22 cells were cultured in DMEM supplemented with 10% foetal bovine serum, 2 mM L-glutamine, 0.4 μg/ml hydrocortisone and 1% penicillin/streptomycin. Cells were routinely checked and confirmed mycoplasma-negative by qPCR (Mycoplasmacheck, Eurofins Genomics) upon thawing and were subsequently used for experiments within two to three passages.

## qRT-PCR

RNA purification was performed using the Monarch Total RNA Miniprep Kit (New England Biolabs) and on-column DNase digestion. cDNA was synthesised from RNA using LunaScript®

Reverse Transcriptase (RT) SuperMix Kit (New England Biolabs). Gene-specific primers were synthesised by IDT and are shown in Table 1. The qRT-PCR primers for *APOBEC3A*, *APOBEC3B* and TATA-binding protein (*TBP*) were published previously (Refsland et al, 2010), and the remaining qRT-PCR primers were designed using OriGene's qPCR primer design tool (https://www.origene.com). All real-time PCR reactions were performed using duplicate technical repeats on a QuantStudio Real-Time PCR system (Applied Biosystems) with amplification using PowerUp™ SYBR™ Green Master Mix for qPCR (Applied Biosystems). The thermal cycling conditions were at 50 °C for 2 min, followed by an initial denaturation step at 95 °C for 10 min, 40 cycles at 95 °C for 15 s and 60 °C for 1 min, followed by 95 °C for 1 min, 60 °C for 1 min and 95 °C for 1 s. Standard curves for *APOBEC3A* and *APOBEC3B* were derived using plasmids pRH3097-A3A (R.S.H lab) and pCMV4-APOBEC3B (a kind gift from Prof Mike Malim, Kings College London, UK) respectively. Standards for all other qRT-PCR target genes were constructed by cloning PCR amplicons generated from a NIKS cDNA library into pCR™ Blunt II-TOPO™ using the Zero Blunt™ TOPO™ PCR cloning kit (Thermo Fisher) as per the manufacturer's instructions. PCR was conducted using the KAPA HiFi 2X MasterMix (Roche) according to the manufacturer's instructions with 10 ng input cDNA, and for all target genes except *TBP* (for which additional primers are listed in Table 1), amplicons were generated using the qRT-PCR primers. All plasmids are available from the authors upon request.

## siRNA transfections

Silencer select small interfering RNAs (siRNA) were purchased from Thermo Fisher Scientific (Negative control (NC#1) Cat No 4390843; GRHL3 (#1) Cat No s33753; GRHL3 (#2) Cat No s33754). NIKS were plated in 6-well plates at a density of $2 \times 10^5$ cells/well and BICR6 and BICR22 at $1.5 \times 10^5$ cells/well, with $1 \times 10^5$ feeder cells/well and were transfected with 2 nM of siRNA using 2 uL of Lipofectamine RNAiMAX (Thermo Fisher Scientific) per well according to the manufacturer's instructions (reverse transfection method). Transfection complexes were removed after 24 h, followed by (NIKS) 48 h recovery period and 24 h treatment with 100 nM afatinib (Fisher Scientific #16463748) or removal of EGF and insulin and reduction to 0.5% FBS for 48 h to induce differentiation, or 3 h treatment with 100 ng/ml PMA (Stem Cell Technologies #74042). BICR6 and BICR22 cells were allowed to recover for 42 h post-transfection with a media change 18 h prior to cell collection.

## APOBEC3A KO NIKS

*APOBEC3A* knockout NIKS clones were generated by using CRISPR-Cas9 to insert the sequence tagttagttag (to terminate translation in all three reading frames), followed by the bovine polyadenylation signal (bpA) at the end of exon 1, to generate an allele in which an mRNA encoding only the first seven amino acids of APOBEC3A is produced (the gene targeting strategy is summarised in Appendix Fig. S14A). Following the split nickase CRISPR method for increased specificity (Cong et al, 2013), single guide (sg)RNAs were designed by entering a 200 base-pair genomic sequence surrounding the *APOBEC3A* start codon into http://crispr.mit.edu/. Oligonucleotides (Integrated DNA Technologies)

encoding the sgRNAs (Table 1) were annealed and cloned into pX335-U6-Chimeric_BB-CBh-hSpCas9n(D10A), a gift from Feng Zhang (Addgene plasmid # 42335; http://n2t.net/addgene:42335; RRID:Addgene_42335) using restriction enzyme Bbs I (NEB as per the Zhang lab protocol (https://www.addgene.org/crispr/zhang/). A targeting construct was generated using NEBuilder® HiFi DNA assembly to combine 1 kb 5′ and 3′ homology arms, the translation/transcription terminator sequence and a PGK-Puro-ΔTK cassette (Chen and Bradley, 2000) flanked by FRT sites into a pBluescript plasmid backbone (Appendix Fig. S15A). The 5′ homology arm contained point mutations at the CRISPR target site to prevent re-cutting following successful recombination into the locus. 5′ and 3′ homology arms were synthesised as g-blocks (Integrated DNA Technologies), and the pBluescript backbone and FRT-Puro-ΔTK-FRT cassette (a kind gift from Dr Su Kit Chew) were amplified by HiFi PCR using primers containing adaptors for NEBuilder assembly. The targeting construct was verified by Sanger sequencing and co-transfected into NIKS, together with pX335 constructs containing the left and right sgRNAs using Fugene HD transfection reagent (Promega), as per the manufacturer's recommendations. Transfected NIKS were selected using puromycin (0.75 µg/ml) for 48 h, and following PCR genotyping to detect successful targeting, single cell clones were generated by plating NIKS at limiting dilution in 96 well plates, with each well containing $1 \times 10^3$ mitomycin C-treated 3T3-J2 feeder cells. Feeder cells were replenished weekly as clones were expanded. Clones were screened by PCR genotyping to identify homozygous knockouts and transfected with GFP-Flippase (generated by cloning mouse codon-optimised Flippase (Flpo), a gift from Philippe Soriano (Addgene plasmid #13792; http://n2t.net/addgene:13792; RRID:Addgene_13792) into pEGFP-N1 (Clontech)). Transfected cells were isolated by fluorescence-activated cell sorting and PCR genotyping was conducted to confirm removal of the FRT-flanked PGK-Puro-ΔTK cassette. Loss of *APOBEC3A* expression and retention of *APOBEC3B*, *APOBEC3C* and *APOBEC3F* expression in knockout clones (C5, F6, and G6) was confirmed by reverse transcriptase PCR of PMA-stimulated NIKS (100 ng/ml/24 h) using the qRT-PCR primers listed in Table 1 (Appendix Fig S15B). *APOBEC3D*, *APOBEC3G* and *APOBEC3H* were undetectable in wild-type or *APOBEC3A* KO NIKS by endpoint RT-PCR. To test the specificity of anti-APOBEC3A clone UMN-13 in immunohistochemistry, formalin-fixed, paraffin-embedded cell blocks were made by embedding wild-type and *APOBEC3A* KO (clone C5) NIKS cell pellets in agarose following 24 h treatment with 100 ng/ml PMA).

### DDOST RNA editing assay

DDOST editing at C558 was measured as described previously (Oh and Buisson, 2022) using 250 ng input RNA for cDNA synthesis. Digital PCR was conducted using the Absolute-Q instrument (Thermo Fisher), with 1 uL of 1:4-diluted cDNA.

### Statistical analysis

Statistical analysis was performed using GraphPad Prism for qRT-PCR and digital PCR experiments and R (version 4.2.0) for single-cell RNAseq analysis. All statistical tests are stated in the figure legends and/or related Methods sections. All statistical tests were two-tailed unless otherwise stated.

## Data availability

scRNA-seq data for epithelial cells from the Southampton dataset from ten HNSCC cases and seven matched healthy tonsil samples) are available here: https://doi.org/10.5281/zenodo.13742281. Validation/external scRNA-seq datasets are available for healthy lung and oesophagus (Madissoon et al, 2019) at Human Cell Atlas Data Coordination Platform and NCBI BIOPROJECT accession code PRJEB31843, and the following at the Gene Expression Omnibus (GEO): breast (GSE176078) (Wu et al, 2021); healthy skin (GSE130973) (Solé-Boldo et al, 2020); bladder squamous cell carcinoma (GSE190888) (Luo et al, 2022); head and neck squamous cell carcinoma (GSE16469082 and GSE18222781); lung carcinoma (GSE131907, GSE136246, GSE148071, GSE153935, GSE127465, GSE119911) (Prazanowska and Lim, 2023) collected at https://doi.org/10.6084/m9.figshare.c.6222221.v3; oesophageal squamous cell carcinoma (GSE160269) (Zhang et al, 2021) Cancer cell line data were obtained at https://discover.nci.nih.gov/rsconnect/cellminercdb/ (accessed 31/07/2023). ChIP-seq data were obtained from the Cistrome Data Browser (http://cistrome.org/db/#/). *APOBEC3A* KO NIKS clones and the plasmids used to generate them are available from the authors upon request.

The source data of this paper are collected in the following database record: biostudies:S-SCDT-10_1038-S44318-024-00298-9.

## Peer review information

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

## Acknowledgements

This work was supported by funding to TRF from the UK Research and Innovation Biotechnology and Biosciences Research Council (BB/V010271/2) and the Rosetrees Trust (M229, PhD2020\100002, CF-2021-2\101), to GJT from Cancer Research UK (DRCNPG-Jun22\100004) and to RSH from the National Cancer Institute (NCI P01-CA234228) and a Recruitment of Established Investigators Award from the Cancer Prevention and Research Institute of Texas (CPRIT RR220053). RSH is an Investigator of the Howard Hughes Medical Institute and the Ewing Halsell President's Council Distinguished Chair at the University of Texas Health San Antonio. P.P. was supported by a CASE studentship from the BBSRC South Coast Biosciences Doctoral Training Partnership (BB/T008768/1) and AstraZeneca. DA, Academic Clinical Fellow (ACF-2023-26-009), is funded by Health Education England (HEE)/NIHR for this research project. The views expressed in this publication are those of the authors and not necessarily those of the NIHR, NHS or the UK Department of Health and Social Care. We would like to thank the Research Histology Group, Dept of Cellular Pathology, University Hospital Southampton, the Faculty of Medicine Tissue Bank, University of Southampton, and Bio-R Bioinformatics Research Facility, University of Southampton. The authors would also like to thank Holly McCarron for her assistance with the design and realisation of the synopsis figure.

## Author contributions

**Nicola J Smith**: Data curation; Formal analysis; Investigation; Methodology; Writing—original draft. **Ian Reddin**: Data curation; Formal analysis; Investigation; Visualisation; Methodology; Writing—original draft.
**Paige Policelli**: Data curation; Formal analysis; Investigation; Methodology; Writing—review and editing. **Sunwoo Oh**: Investigation.
**Nur Zainal**: Investigation; Visualisation. **Emma Howes**: Investigation.
**Benjamin Jenkins**: Data curation; Formal analysis; Investigation; Visualisation. **Ian Tracy**: Investigation. **Mark Edmond**: Investigation.
**Benjamin Sharpe**: Investigation; Visualisation. **Damian Amendra**: Investigation. **Ke Zheng**: Investigation. **Nagayasu Egawa**: Investigation. **John Doorbar**: Supervision; Investigation. **Anjali Rao**: Formal analysis; Investigation.
**Sangeetha Mahadevan**: Formal analysis; Investigation. **Michael A Carpenter**: Resources. **Reuben S Harris**: Resources; Funding acquisition; Writing—review and editing. **Simak Ali**: Investigation; Writing—review and editing.
**Christopher Hanley**: Formal analysis; Investigation. **Rémi Buisson**: Supervision; Investigation; Writing—review and editing. **Emma King**: Conceptualisation; Supervision; Investigation. **Gareth J Thomas**: Conceptualisation; Formal analysis; Supervision; Funding acquisition; Investigation; Writing—review and editing. **Tim R Fenton**: Conceptualisation; Formal analysis; Supervision; Funding acquisition; Investigation; Methodology; Writing—original draft; Project administration.

Source data underlying figure panels in this paper may have individual authorship assigned. Where available, figure panel/source data authorship is listed in the following database record: biostudies:S-SCDT-10_1038-S44318-024-00298-9.

## Disclosure and competing interests statement

TRF is an advisory board member of and holds stock options in APOBEC Discovery Ltd.

# Expanded View Figures

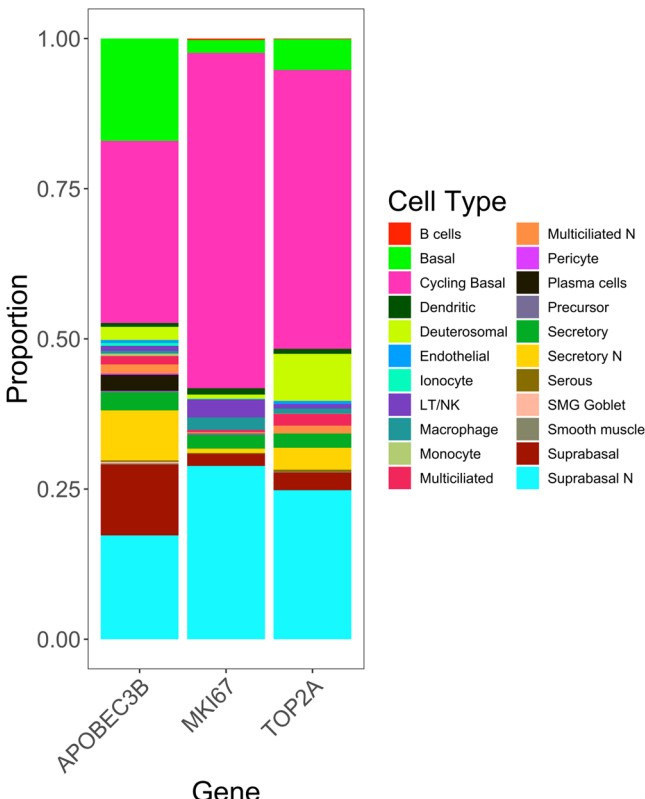

**Figure EV1.** ***APOBEC3B* is predominantly co-expressed with proliferation markers *MKI67* and *MCM7* in cycling basal and suprabasal cells of the healthy human airway.**

Stacked barplot showing the proportion of each cell type in the single cell atlas of healthy human airways from Deprez et al that express each selected gene.

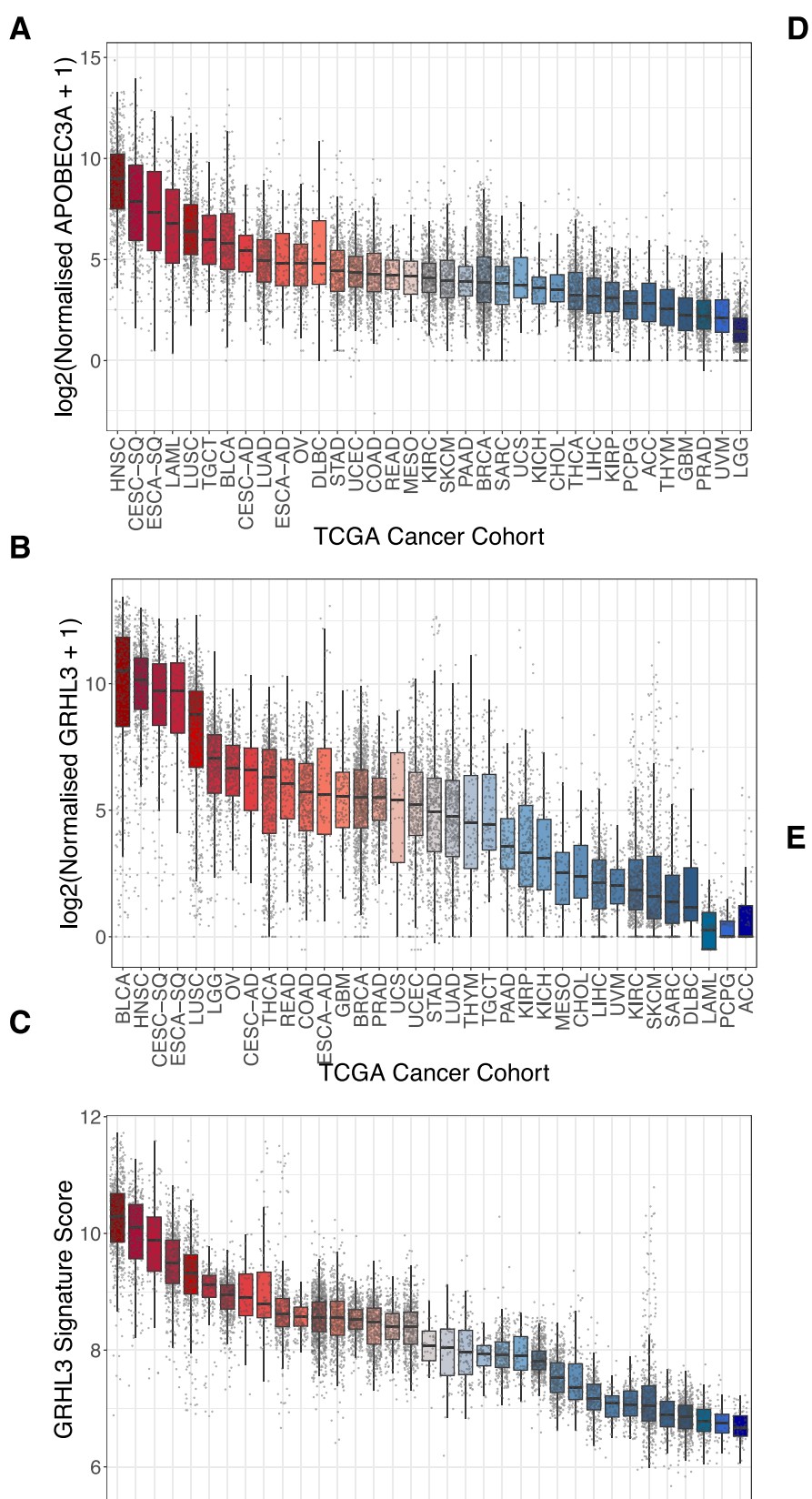

**D**

| TCGA Cancer Type | n | Spearman Correlation | Adjusted p-value |
|---|---|---|---|
| ESCA-SQ | 99 | 0.65 | 2.48E-12 |
| CESC-SQ | 255 | 0.62 | 2.10E-27 |
| CESC-AD | 54 | 0.51 | 2.70E-04 |
| ESCA-AD | 97 | 0.48 | 2.17E-06 |
| THYM | 122 | 0.47 | 2.15E-07 |
| HNSC | 566 | 0.43 | 2.31E-25 |
| THCA | 572 | 0.40 | 3.45E-22 |
| LUSC | 552 | 0.32 | 9.80E-14 |
| PRAD | 550 | 0.20 | 8.56E-06 |
| BRCA | 1,215 | 0.19 | 4.07E-10 |
| CHOL | 45 | 0.15 | 3.94E-01 |
| OV | 308 | 0.14 | 3.29E-02 |
| BLCA | 427 | 0.12 | 3.75E-02 |
| KICH | 91 | 0.10 | 4.36E-01 |
| UCEC | 555 | 0.09 | 7.62E-02 |
| PAAD | 183 | 0.07 | 3.94E-01 |
| SKCM | 473 | 0.06 | 2.59E-01 |
| STAD | 450 | 0.06 | 3.03E-01 |
| LUAD | 576 | 0.06 | 2.59E-01 |
| LAML | 173 | 0.00 | 9.64E-01 |
| KIRP | 323 | 0.00 | 9.64E-01 |
| DLBC | 48 | -0.01 | 9.64E-01 |
| LIHC | 423 | -0.01 | 8.49E-01 |
| ACC | 79 | -0.03 | 8.49E-01 |
| LGG | 529 | -0.12 | 2.05E-02 |
| SARC | 265 | -0.12 | 1.06E-01 |
| GBM | 166 | -0.12 | 1.96E-01 |
| TGCT | 139 | -0.13 | 1.96E-01 |
| PCPG | 187 | -0.13 | 1.30E-01 |
| READ | 170 | -0.15 | 8.67E-02 |
| COAD | 492 | -0.16 | 9.22E-04 |
| KIRC | 606 | -0.16 | 1.75E-04 |
| MESO | 87 | -0.23 | 6.60E-02 |

**E**

| TCGA Cancer Type | n | Spearman Correlation | Adjusted p-value |
|---|---|---|---|
| ESCA-SQ | 99 | 0.71 | 1.2E-15 |
| CESC-SQ | 255 | 0.68 | 3.7E-35 |
| ESCA-AD | 97 | 0.64 | 5.9E-12 |
| CESC-AD | 54 | 0.57 | 9.1E-06 |
| LAML | 173 | 0.56 | 3.3E-15 |
| LUSC | 552 | 0.54 | 3.9E-42 |
| THYM | 122 | 0.54 | 2.9E-10 |
| HNSC | 566 | 0.54 | 4.0E-42 |
| THCA | 572 | 0.53 | 5.9E-41 |
| KICH | 91 | 0.46 | 6.8E-06 |
| BLCA | 427 | 0.45 | 8.6E-22 |
| DLBC | 48 | 0.37 | 1.2E-02 |
| SKCM | 473 | 0.37 | 1.0E-15 |
| SARC | 265 | 0.34 | 2.2E-08 |
| PRAD | 550 | 0.32 | 4.7E-14 |
| PCPG | 187 | 0.31 | 2.9E-05 |
| GBM | 166 | 0.31 | 8.6E-05 |
| LIHC | 423 | 0.30 | 5.7E-10 |
| LGG | 529 | 0.28 | 2.3E-10 |
| STAD | 450 | 0.27 | 1.4E-08 |
| OV | 308 | 0.25 | 1.5E-05 |
| LUAD | 576 | 0.24 | 1.3E-08 |
| COAD | 492 | 0.24 | 1.7E-07 |
| CHOL | 45 | 0.22 | 1.7E-01 |
| KIRP | 323 | 0.22 | 1.1E-04 |
| UCEC | 555 | 0.18 | 1.8E-05 |
| BRCA | 1,215 | 0.17 | 2.0E-09 |
| READ | 170 | 0.16 | 4.6E-02 |
| PAAD | 183 | 0.07 | 3.3E-01 |
| MESO | 87 | 0.04 | 7.5E-01 |
| TGCT | 139 | 0.01 | 9.0E-01 |
| KIRC | 606 | -0.07 | 8.6E-02 |
| ACC | 79 | -0.16 | 1.8E-01 |

◄ **Figure EV2. *APOBEC3A* and *GRHL3* expression by cancer type in TCGA bulk RNA-seq data.**

(A) Boxplot ranked from highest median (left) to lowest median (right) for *APOBEC3A* gene expression. (B) Boxplot ranked from highest median (left) to lowest median (right) for *GRHL3* gene expression. (C) Boxplot ranked from highest median (left) to lowest median (right) for GRHL3 signature score based on the mean expression of 127 GRHL3 target genes identified in scRNA-seq SCENIC analysis. Each dot represents an individual tumour sample. (D) Spearman correlation coefficients for *APOBEC3A* and *GRHL3* expression in TCGA RNA-seq data. (E) Spearman correlation coefficients for *APOBEC3A* expression and GRHL3 signature score in TCGA RNA-seq data. ESCA and CESC were stratified into squamous cell (SQ) and adeno- (AD) carcinomas. For all boxplots, the top, middle and bottom lines of the boxplot represent the upper quartile (Q3), median, and lower quartile (Q1), respectively. The maximum value represented by the top whisker represents the highest observed data point within $Q3 + (1.5 \times (Q3 - Q1))$, and the minimum value represented by the bottom whisker represents the lowest situated point within $Q1 - (1.5 \times (Q3 - Q1))$.

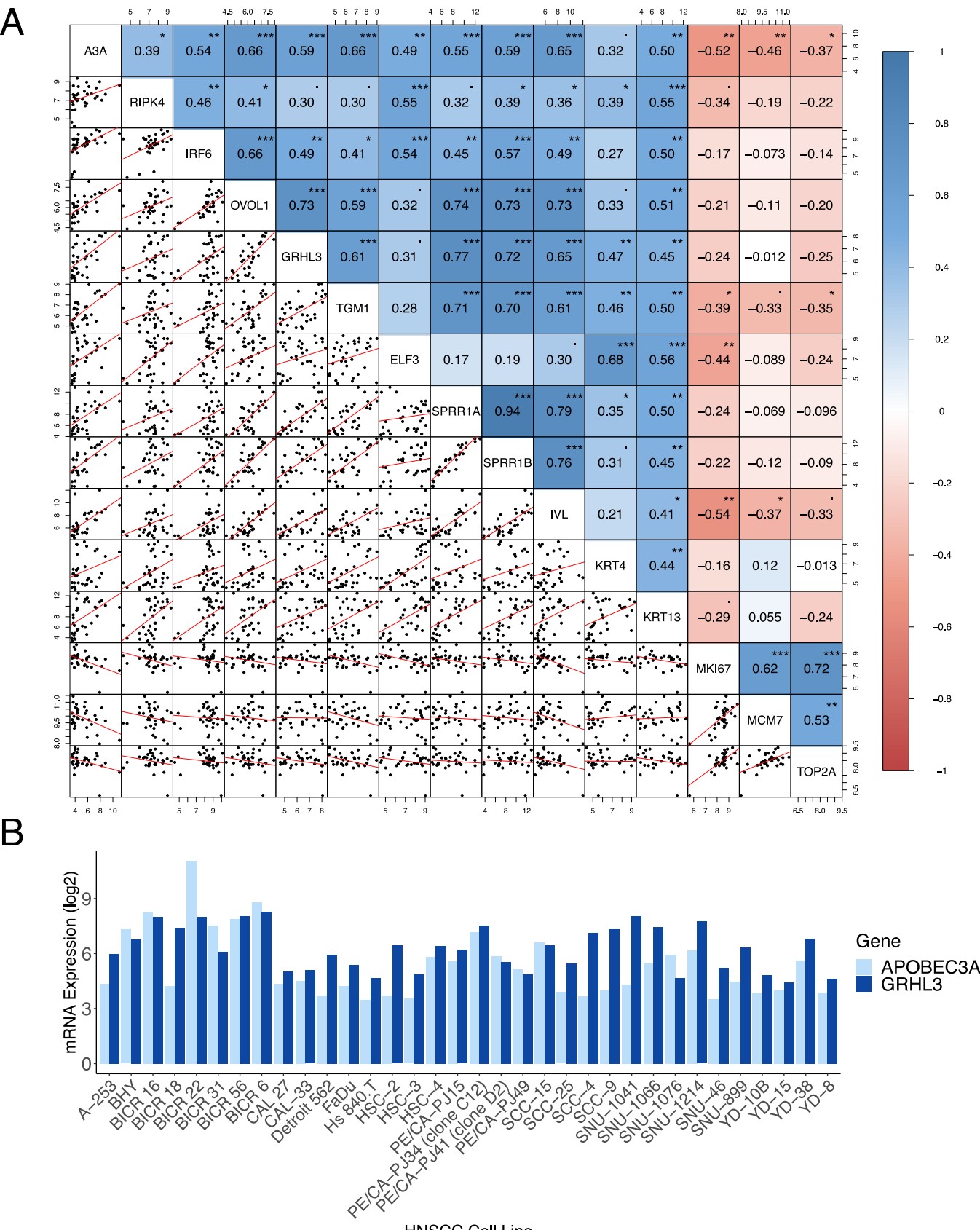

**Figure EV3.  *APOBEC3A* is positively correlated with differentiation marker genes and negatively correlated with proliferation marker genes in HNSCC cell lines.**

(**A**) Spearman's correlation of *APOBEC3A* expression with the expression of differentiation marker genes, proliferation marker genes, and genes of the RIPK4 pathway in 34 HNSCC cell lines from the CCLE. *P* values calculated by *T*-test: \*\*\**P* < 0.0001; \*\**P* < 0.001; \**P* < 0.05; *P* < 0.1. (**B**) Log2 expression levels of *APOBEC3A* and *GRHL3* in the 34 individual HNSCC cell lines from the CCLE. Accompanies Fig. 5.

**APOBEC3A**          **APOBEC3A/B/G**

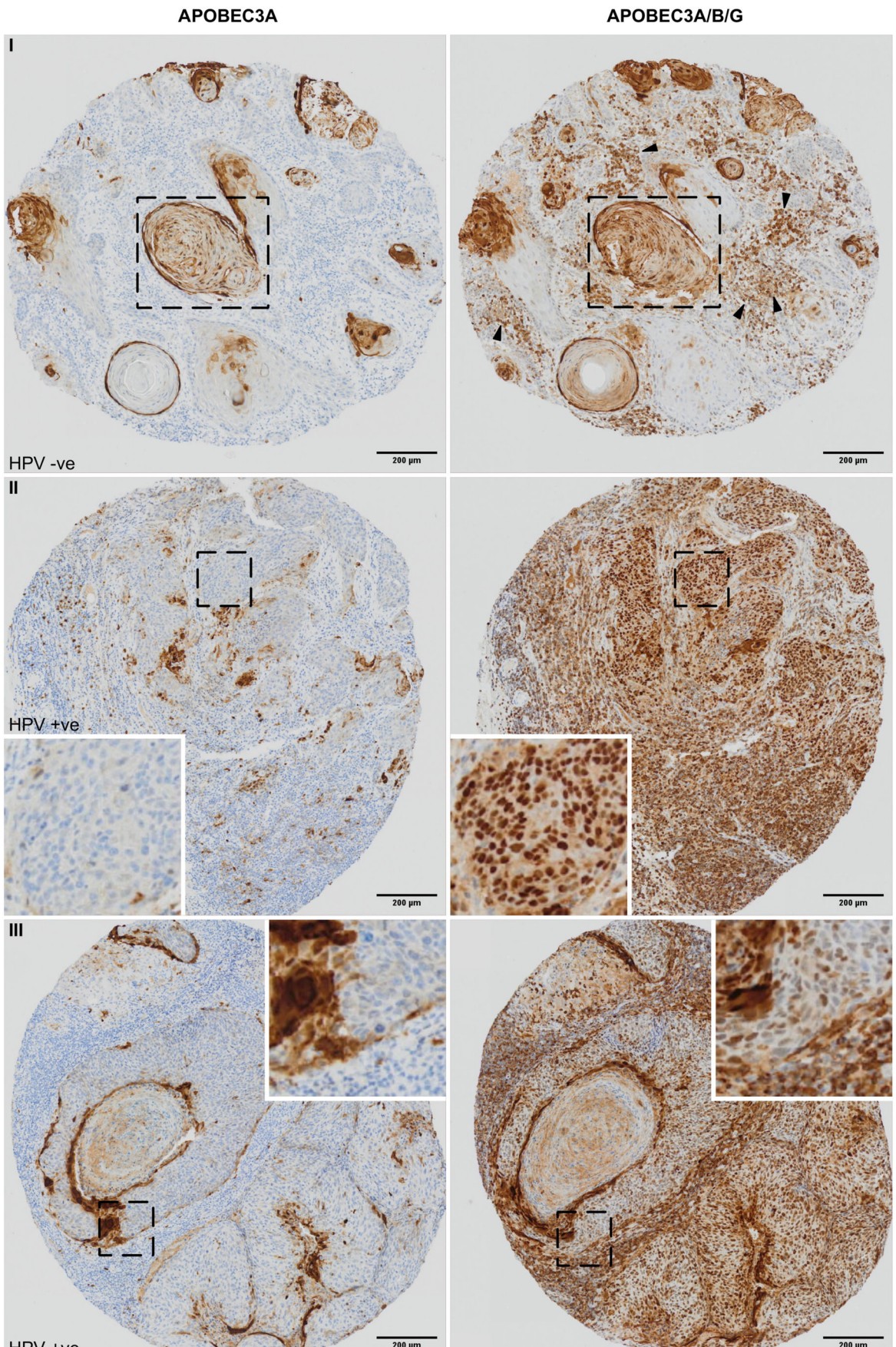

**Figure EV4.  Detection of APOBEC3A and APOBEC3B protein expression in HNSCC.**

Representative images from an HNSCC tissue microarray stained with APOBEC3A-specific (left panels) or APOBEC3A/B/G-specific (right panels) antibodies. Accompanies Fig. 5.

