## [Peer Review File · The EMBO Journal]

Differentiation signals induce APOBEC3A via GRHL3 in squamous epithelia and squamous cell carcinoma

Nicola Smith, Ian Reddin, Paige Policelli, Sunwoo Oh, Nur Zainal, Emma Howes, Benjamin Jenkins, Ian Tracy, Mark Edmond, Benjamin Sharpe, Damian Amendra, Ke Zheng, Nagayasu Egawa, John Doorbar, Anjali Rao, Sangeetha Mahadevan, Michael Carpenter, Reuben Harris, Simak Ali, Christopher Hanley, Rémi Buisson, Emma King, Gareth Thomas, and Tim Fenton

Corresponding author: Tim Fenton (t.r.fenton@soton.ac.uk)

Review Timeline:

Transferred from Review Commons:	11th May 24
Editorial Decision:	22nd May 24
Revision Received:	11th Sep 24
Editorial Decision:	11th Oct 24
Revision Received:	21st Oct 24
Accepted:	24th Oct 24

Editors: Kelly Anderson / Daniel Klimmeck

Transaction Report:

This manuscript was transferred to The EMBO Journal following peer review at Review Commons.

Review #1**1. Evidence, reproducibility and clarity:****Evidence, reproducibility and clarity (Required)******Summary:****

In this study, the authors performed single cell RNA sequencing to study the expression of APOBEC3A (A3A) and APOBEC3B (A3B) across different healthy and malignant tissue samples. The key finding of the study is that while A3B expression was restricted to proliferating cells, A3A was expressed mainly in differentiating normal cells. These results were validated by spatial transcriptomics, IHC and functional experiments. The study identified Grainyhead-like transcription factor 3 (GRHL3) as a transcriptional regulator of A3A in terminally differentiating keratinocytes and SCC tumor cells. GRHL3 was found to correlate with the expression and cytidine deaminase activity of A3A. Overall the study identifies specific cell lineages that express either A3A or A3B in normal tissues and identifies GRHL3 as a previously unknown transcriptional regulator of A3A in normal tissues and certain cancers.

****Major comments:****

1. The authors discuss breast cancers to be rich in A3B, consistent with cancer cell proliferation, however cancers like head and neck and that of the oesophagus appear to express both A3A and A3B. It is unclear from the description whether there are cases where A3A and A3B are coexpressed in cancer. Can the authors determine this from their dataset?
2. In line 130 the authors state "To address this question, the 2,649 epithelial cells derived from normal tonsil were subset from our 21,937 epithelial cell dataset (Figure 2a) and pathway analysis using gene ontology biological processes (GOBP) was performed on the top 100 genes co-expressed with either APOBEC3A or APOBEC3B (Table S3)." This sentence is a bit confusing and looking at 2a, it is impossible to extrapolate what subset they are referring to. This statement and accompanying data could use some clarification for the reader.
3. In line 236-239 authors say, "Furthermore, GRHL3 was the only transcription factor among those whose activity was correlated with APOBEC3A expression that was significantly upregulated in the differentiating compartment of the normal tonsil

epithelium, in which most APOBEC3A-expressing cells were clustered (Figure 4a, b, Figure 1e, Table S6)". However, there is no figure 1e provided.

4. Was a similar peak motif also found in the promoters/enhancers of A3A for IVL and ELF3 that are also downregulated upon GHRL3 depletion? From the data it would appear that A3A is transiently expressed during differentiation and then repressed in terminally differentiated cells, suggesting that some repressors must be subsequently recruited.

5. How specific is GHRL3 expression to keratinocytes? In other words, would this mechanism potentially be relevant in breast or other tissue that exhibits APOBEC mutational signatures?

6. Did the authors identify any transcription factors in spatial scRNA-seq data that correlate with A3B expression?

7. The results heading for Figure 4 states that A3A expression is induced by GRHL3. While the data is consistent with this, it is not actually demonstrated that expression of GRHL3 directly causes A3A induction. The converse is shown in depletion experiments, however I feel that stating A3A expression in differentiation requires GRHL3 or something to that effect more accurately reflects the data in the absence of evidence for a more cause-effect relationship. Similarly, the heading for the next section "GRHL3 drives APOBEC3A expression in HNSCC and ESCC" is based entirely on correlation. I therefore feel that, while likely, this is an overstatement of the actual data presented.

****Minor comments:****

1. APOBEC3G is misspelled in line 368.

2. We did not have access to any supplementary tables for review.

2. Significance:

Significance (Required)

****Significance:****

***General assessment:** The study provides new information about the endogenous regulation of APOBECs, a topic for which there is little knowledge and considerable interest

given their roles in cancer. It also adds additional detail to previous observations that EGFR inhibitors can activate A3A expression.

***Advance:** The authors demonstrate that A3A and A3B are differentially regulated and implicated GRHL3 as a main regulator of A3A in differentiating cells, also identifying a likely promoter element for A3A. Further, they provide evidence that A3B is expressed primarily in cycling cells. These observations raise interesting questions about the role of A3A in cancer. Many cancers exhibit a gradient of differentiation and these results may suggest that this process would lead to A3A expression and subsequent mutations that could lead to more aggressive tumor growth. Additionally, it suggests that there may be a role for A3A expression in differentiating cells under homeostatic conditions.

***Limitations of the study:** The results rely heavily on the analysis of scRNAseq experiments. While these approaches are ideal for identifying cell populations, they are also not as sensitive as other methods for detecting the expression of genes. While some depletion and ChIP experiments are carried out, much of the model relies on correlation rather than direct manipulation of promoter elements etc. While all of the data is consistent with the author's proposals, it is a limitation of the study.

***Audience:** The results will be of interest to a wide audience interested in cancer mutational signatures, cancer etiology, antiviral responses and DNA damage and repair.

***Expertise:** Our expertise is in the mechanisms of the DNA damage response, including to APOBEC activity, and its impact on cancer etiology. We do not have significant experience with the generation or analysis of scRNAseq data.

3. How much time do you estimate the authors will need to complete the suggested revisions:

Estimated time to Complete Revisions (Required)

(Decision Recommendation)

Less than 1 month

4. Review Commons values the work of reviewers and encourages them to get credit for their work. Select 'Yes' below to register your reviewing activity at Web of Science Reviewer Recognition Service (formerly Publons); note that the content of your review will not be visible on Web of Science.

Yes

Review #2

1. Evidence, reproducibility and clarity:

Evidence, reproducibility and clarity (Required)

****Summary:****

Smith and colleagues report novel findings related to regulation of expression of APOBEC3A and APOBEC3B, two candidate cancer mutagens, in both squamous cell cancers and healthy epithelia. Through de novo single cell RNA sequencing and analysis of published data sets, Smith et al find that the GRHL3 pioneer transcription factor regulates APOBEC3A to restrict expression to differentiating/terminally differentiated cells. In contrast, APOBEC3B is expressed in proliferating cells during G2-M phase and expression is associated with mitotic markers.

I have only minor comments/suggestions:

1. Could the authors comment on the difference in scale between A3A and A3B expression in Fig 2e? It seems that A3B is expressed at much lower levels than A3A in tonsils. Is the same true in oesophagus (Fig S1)?
2. NIKS experiments in Fig 3/S2 would benefit from a non-qPCR marker of cell cycle and/or proliferation. While data regarding A3A/B expression correlated with proliferation/cycle markers are robust, it would be nice to see complementary experiments to show proliferation or cell cycle in this system.
3. Does GRHL3 mediate A3A expression in non-afatinib induction conditions? In other words, if one of the other systems used in Fig 3 were subject to GRHL3 depletion, would A3A expression be inhibited?
4. In Fig 4f the GRHL3 transcription factor is identified from in silico data at two regulatory sites proximal to A3A. WDR5 is presumably recruited by GRHL3 and is found at similar locations. Do the authors think that the same chromatin regulation of A3A occurs in all cells?

5. NFκB has been reported as a transcriptional mediator of APOBEC3A expression in two recent publications (Isozaki, et al PMID 37407818, Oh, et al PMID 34389714). However, NFκB does not appear in the list of transcription factors whose activity correlated with APOBEC3A expression in keratinocytes (Fig 4a). Can the authors comment on where NFκB might fit into the GRHL3-APOBEC3A axis?

****Minor clarifications:****

- There is a typo on line 239 where Figure 1e is called out, but I believe the authors meant 2e.
- Supplemental Fig 6d - the heatmap legend does not have color variation, which seems to be an error.
- In Fig 5b, was qPCR performed on sub-confluent cultures (as in Fig S9)?

2. Significance:

Significance (Required)

This manuscript tackles an essential problem in cancer mutagenesis, which is the normal and aberrant regulation of the APOBEC enzymes. The data are robust and the manuscript is well written. The switch between A3B expression in cycling cells and A3A expression in differentiating cells fills a gap in knowledge that makes sense of many prior published findings. The impact of this manuscript is substantial and sets up the field for further analysis of APOBEC regulation and dysregulation in both healthy and malignant tissues beyond keratinocytes and squamous cell tumors.

3. How much time do you estimate the authors will need to complete the suggested revisions:

Estimated time to Complete Revisions (Required)

(Decision Recommendation)

Between 1 and 3 months

4. Review Commons values the work of reviewers and encourages them to get credit for their work. Select 'Yes' below to register your reviewing activity at Web of Science Reviewer Recognition Service (formerly Publons); note that the content of your review will not be visible on Web of Science.

No

Review #3

1. Evidence, reproducibility and clarity:

Evidence, reproducibility and clarity (Required)

This study explores APOBEC3A and APOBEC3B expression, regulation, and their implications for normal cellular function versus potential cancer-related dysregulation. The emphasis is on squamous cell carcinomas of head and neck (HNSCC) and esophagus (ESCC).

****Approach:****

A robust, multi-pronged approach is used, including single-cell RNA sequencing (scRNA-Seq), spatial transcriptomics, antibody validation, and computational analysis.

****Findings:****

- GRHL3 emerges as a central regulator of APOBEC3A, particularly in keratinocytes. APOBEC3B expression aligns with cell division patterns, supporting its role in DNA replication-linked mutagenicity.
- The correlation of GRHL3, APOBEC3A, and cellular replication status in HNSCC and ESCC suggests a potential oncogenic mechanism.

****Conclusions:****

APOBEC3A appears to have a tightly controlled role in normal skin cell differentiation with implications for skin health. GRHL3, or its associated signaling networks, may represent promising therapeutic targets in cancers driven by APOBEC3-related mutations.

2. Significance:

Significance (Required)

The link between GRHL3 and APOBEC3A discovered in the manuscript is compelling, however, further validation will strengthen the study. Directly manipulating GRHL3 in cancer models to observe changes in APOBEC-induced mutations would support the conclusion that it is a primary driver of this mechanism.

****GRHL3 Specificity:****

Tissue Context: It's important to determine if GRHL3-mediated APOBEC3A activation is unique to cancers explored here, or a broader principle in epithelial malignancies.

Regulation: Clarify if the link is unique to dysregulated cancer cells, or might reflect normal cellular processes amplified in oncogenesis.

Differentiation Focus: This study uniquely emphasises APOBEC3A's potential in differentiation, an angle to explicitly highlight as it may have far-reaching implications beyond cancer understanding.

Therapeutic Breadth: Explore if the results are likely limited to the studied cancers, or have potential application across a wider range of epithelial cancers.

This study presents a significant contribution to our understanding of APOBEC3A and APOBEC3B function, with particular emphasis on their roles in both normal cellular processes and cancer development. The methodological rigor, including the multi-pronged approach and focus on cellular context, provides a detailed and insightful view. The identification of GRHL3 as a central regulator of APOBEC3A presents a promising novel therapeutic target for combating APOBEC-related mutations in cancer. Furthermore, by highlighting APOBEC3A's potential role in cellular differentiation, the study offers a new perspective on these enzymes that could have broader implications for understanding skin health and other physiological processes. While further experimental validation would definitively solidify the causative link between GRHL3 and APOBEC-induced mutagenesis, this work meaningfully advances the field. Its conceptual insights into episodic mutation and potential dysregulation of differentiation in cancer have the potential to influence future research directions and cancer treatment strategies.

3. How much time do you estimate the authors will need to complete the suggested revisions:

Estimated time to Complete Revisions (Required)

(Decision Recommendation)

Between 1 and 3 months

4. Review Commons values the work of reviewers and encourages them to get credit for their work. Select 'Yes' below to register your reviewing activity at Web of Science

Reviewer Recognition Service (formerly Publons); note that the content of your review will not be visible on Web of Science.

Yes

Revision Plan

Manuscript number: RC-2024-02419

Corresponding author(s): Tim Fenton

[The “revision plan” should delineate the revisions that authors intend to carry out in response to the points raised by the referees. It also provides the authors with the opportunity to explain their view of the paper and of the referee reports.]

The document is important for the editors of affiliate journals when they make a first decision on the transferred manuscript. It will also be useful to readers of the reprint and help them to obtain a balanced view of the paper.

*If you wish to submit a full revision, please use our "Full Revision" template. **It is important to use the appropriate template to clearly inform the editors of your intentions.**]*

1. General Statements [optional]

This section is optional. Insert here any general statements you wish to make about the goal of the study or about the reviews.

Despite their prominent role in cancer and their emergence as cancer drug targets, remarkable little is known regarding the regulation of *APOBEC3A* and *APOBEC3B* gene expression. This knowledge gap has led to considerable debate in the field and confusion as to whether efforts should be made to selectively target either enzyme in different contexts / cancer types (Petljak et al *Nature Genetics* **54**, pages 1599–1608 (2022)).

In this study, we used a combination of single cell RNA sequencing, spatial transcriptomics and immunohistochemical analysis of human tissue samples, supported by experiments in cultured keratinocytes and cancer cell lines to enhance our understanding of *APOBEC3A* and *APOBEC3B* regulation.

We thank the reviewers for their appraisal of our manuscript and were pleased to note their positive comments regarding the robustness of our data and of the significance of our findings to a wide audience.

2. Description of the planned revisions

Insert here a point-by-point reply that explains what revisions, additional experimentations and analyses are planned to address the points raised by the referees.

Reviewer #1

Revision Plan

Comment 1: The authors discuss breast cancers to be rich in A3B, consistent with cancer cell proliferation, however cancers like head and neck and that of the oesophagus appear to express both A3A and A3B. It is unclear from the description whether there are cases where A3A and A3B are coexpressed in cancer. Can the authors determine this from their dataset?

We do indeed see a minority of A3A expressing cells that also express A3B, both in normal and cancer samples. We speculate that at least in normal epithelium, these are cells that may have exited cell cycle in G2/M and initiated differentiation, as has been described previously for keratinocytes. In cancers, co-expression may (also) occur due to dysregulation of either A3A or A3B expression, for example where we see A3A expression in a minority population of dividing cells in some HNSCC cases (Figure 5f). However, this population is so small that we don't currently have enough cells available to conduct a meaningful analysis (e.g. to determine which other genes are co-expressed / pathways enriched) in these A3A/A3B co-expressing cells. We will endeavour to find additional published scRNA-seq datasets with sufficient numbers of A3A/A3B co-expressing cells to address this question if possible.

Comment 4: Was a similar peak motif also found in the promoters/enhancers of A3A for IVL and ELF3 that are also downregulated upon GHRL3 depletion? From the data it would appear that A3A is transiently expressed during differentiation and then repressed in terminally differentiated cells, suggesting that some repressors must be subsequently recruited.

We will inspect GRHL3 ChIP-seq data for binding peaks in the IVL and ELF3 promoters and will include other GRHL3 target genes including TGM1. In cases where peaks are present, we will search for the putative GRHL3 binding motifs we identified in APOBEC3A.

We agree that APOBEC3A expression could be repressed in terminally differentiated cells and will inspect our SCENIC analysis for the presence of any transcriptional repressors that show increased activity in terminally differentiated cells, and which include APOBEC3A among their regulon (set of predicted target genes).

It will be difficult for us to validate such a repressor in our cell culture system as the keratinocytes do not reach a fully differentiated state but the identification of such a candidate factor would be of interest and would likely stimulate further investigation.

Comment 5: How specific is GHRL3 expression to keratinocytes? In other words, would this mechanism potentially be relevant in breast or other tissue that exhibits APOBEC mutational signatures?

We agree that establishing this would give valuable insight into how widely this mechanism may operate across human cancers in which APOBEC signature mutations are observed. In our analysis of scRNA-seq datasets from cancer types including head/neck, oesophagus, lung, breast and bladder, we only observed APOBEC3A expression in a sufficient number of cells to permit SCENIC analysis (to identify potential transcription factors) in head/neck and

Revision Plan

oesophagus. However, having identified GRHL3 as a potential APOBEC3A regulator from the HNSCC and ESCC datasets, we will attempt to determine the extent to which GRHL3 and APOBEC3A are co-expressed in the scRNA-seq data from these additional cancer types.

We will also assess GRHL3 and APOBEC3A expression in publicly available datasets including GTEX and the Human Protein Atlas (both contain scRNA-seq and bulk RNA-seq from normal tissues), The Cancer Genome Atlas project (bulk RNA-seq from cancer samples) and the Cancer Cell Line Encyclopaedia (bulk RNA-seq from cancer cell lines). Initial inspection of these datasets suggests oral and oesophageal mucosa are indeed sites of particularly high APOBEC3A expression, and that GRHL3 is also expressed at high levels in these tissues. Initial analysis of TCGA data suggests strong correlation between GRHL3 and APOBEC3A expression in squamous cell carcinomas of the head and neck, oesophagus, lung and cervix but not in breast cancers, where GRHL3 and APOBEC3A expression is considerably lower. We propose to conduct further analysis, including stratification of the cervical and lung cancer data into squamous cell carcinomas and adenocarcinomas.

Reviewer #2

Comment 2: NIKS experiments in Fig 3/S2 would benefit from a non-qPCR marker of cell cycle and/or proliferation. While data regarding A3A/B expression correlated with proliferation/cycle markers are robust, it would be nice to see complementary experiments to show proliferation or cell cycle in this system.

We will perform flow cytometry on fixed and Propidium Iodide-stained cells collected under these conditions to provide a complementary and more direct assessment of cell cycle status in these experiments. These results will be added to Figure 3 and/or Figure S2.

Comment 3: Does GRHL3 mediate A3A expression in non-afatinib induction conditions? In other words, if one of the other systems used in Fig 3 were subject to GRHL3 depletion, would A3A expression be inhibited?

We are currently performing GRHL3 knockdown experiments using these additional methods of inducing differentiation / A3A expression and include the results in a new supplementary figure.

Comment 4: In Fig 4f the GRHL3 transcription factor is identified from in silico data at two regulatory sites proximal to A3A. WDR5 is presumably recruited by GRHL3 and is found at similar locations. Do the authors think that the same chromatin regulation of A3A occurs in all cells?

This is an interesting point. Apart from keratinocytes, A3A is also highly expressed in neutrophils and monocytes, and remains interferon-inducible in macrophage. Based on their expression patterns, it is unlikely that GRHL3 or other Grainyhead family transcription factors are involved in regulating A3A in these cell types. We will analyse publicly available ChIP-seq

Revision Plan

data to determine the histone modification status of the relevant regulatory sites across different cell types.

Comment 5: NFκB has been reported as a transcriptional mediator of APOBEC3A expression in two recent publications (Isozaki, et al PMID 37407818, Oh, et al PMID 34389714). However, NFκB does not appear in the list of transcription factors whose activity correlated with APOBEC3A expression in keratinocytes (Fig 4a). Can the authors comment on where NFκB might fit into the GRHL3-APOBEC3A axis?

As the reviewer points out, we do not see a positive correlation between inferred RELA activity and APOBEC3A expression in scRNA-seq data, at least from normal tonsil. We note that in the study by Oh et al cited by the reviewer, RELA is required for APOBEC3A induction by replication stress in cancer cell lines and we do see positive correlations between predicted RELA activity and APOBEC3A expression in some of the scRNA-seq datasets from tumours (particularly so in oesophageal SCC). GRHL3 has been described as a pioneer factor (Jacobs et al *Nat. Genet.* **50**, 1011–1020 (2018); a transcription factor that can bind to regions of closed chromatin and recruit factors (in the case of GRHL3, the WDR5-containing trithorax H3K4 methyltransferase complex) required for the initial priming of a gene for activation by additional transcription factors. We hypothesize that GRHL3 may therefore be necessary for the activation of APOBEC3A expression by other transcription factors such as NFκB and STAT2.

We propose to conduct transfection experiments in readily transfectable cell lines that do not express APOBEC3A (HeLa, HEK293), to determine whether GRHL3 or RELA overexpression alone or in combination is sufficient to induce APOBEC3A expression. We will also conduct siRNA experiments in which we will target GRHL3 and RELA alone or in combination, to test whether RELA is required for the GRHL3-dependent APOBEC3A expression we observe in cultured keratinocytes.

Reviewer #3

GRHL3 Specificity:

Tissue Context: It's important to determine if GRHL3-mediated APOBEC3A activation is unique to cancers explored here, or a broader principle in epithelial malignancies.

See response to Reviewer #1, comment 5 (directly above).

Therapeutic Breadth: Explore if the results are likely limited to the studied cancers, or have potential application across a wider range of epithelial cancers.

This should also be covered by the investigations we have proposed in response to the point concerning GRHL3 specificity (tissue context) – see response to Reviewer #1, comment 5.

Revision Plan

3. Description of the revisions that have already been incorporated in the transferred manuscript

Please insert a point-by-point reply describing the revisions that were already carried out and included in the transferred manuscript. If no revisions have been carried out yet, please leave this section empty.

All revisions that have been incorporated in revised manuscript are displayed in blue text.

Reviewer #1

Comment 2: In line 130 the authors state "To address this question, the 2,649 epithelial cells derived from normal tonsil were subset from our 21,937 epithelial cell dataset (Figure 2a) and pathway analysis using gene ontology biological processes (GOBP) was performed on the top 100 genes co-expressed with either APOBEC3A or APOBEC3B (Table S3)." This sentence is a bit confusing and looking at 2a, it is impossible to extrapolate what subset they are referring to. This statement and accompanying data could use some clarification for the reader.

We have clarified this section of the text to read as follows (lines 131-135): "To address this question, we identified the top 100 genes co-expressed with either APOBEC3A or APOBEC3B (Table S3) in the 2,649 epithelial cells from our healthy tonsil samples (Figure 2a, teal) and used pathway analysis to examine the gene ontology biological processes (GOBP) enriched among each geneset (Figure 2b)."

We have also amended the legend for Figure 2 as follows: "(a) UMAP projection of 21,937 epithelial cells from oropharyngeal squamous cell carcinoma samples (n = 10; 19,314 cells (pink)), and matched normal tonsil (n = 7; 2,649 cells (teal)). b) GOBP terms enriched amongst the sets of 100 genes that were co-expressed with either APOBEC3A or APOBEC3B in the 2,469 epithelial cells from normal tonsil."

Comment 3: In line 236-239 authors say, "Furthermore, GRHL3 was the only transcription factor among those whose activity was correlated with APOBEC3A expression that was significantly upregulated in the differentiating compartment of the normal tonsil epithelium, in which most APOBEC3A-expressing cells were clustered (Figure 4a, b, Figure 1e, Table S6)". However, there is no figure 1e provided.

The reference to Figure 1e was a mistake. We have corrected this to Figure 2e.

Comment 6: Did the authors identify any transcription factors in spatial scRNA-seq data that correlate with A3B expression?

We have added the A3B results of our SCENIC analysis to Table S5 and have added the following to the results text (lines 272-276): "Although no single transcription factor activity

Revision Plan

displayed a consistently strong correlation with *APOBEC3B* expression (Table S5, possibly due to the lower number of A3B-expressing cells in several of the datasets), correlations with E2F family transcription factors are consistent with previous reports linking repressive E2F-containing complexes to *APOBEC3B* regulation (Periyasamy et al *Nucleic Acids Research* 45(19):11056-11069 (2017); Roelofs et al *Elife* 9:e61287 (2020) and with its expression in proliferating cells.”

Comment 7: The results heading for Figure 4 states that A3A expression is induced by GRHL3. While the data is consistent with this, it is not actually demonstrated that expression of GRHL3 directly causes A3A induction. The converse is shown in depletion experiments, however I feel that stating A3A expression in differentiation requires GRHL3 or something to that effect more accurately reflects the data in the absence of evidence for a more cause-effect relationship. Similarly, the heading for the next section "GRHL3 drives APOBEC3A expression in HNSCC and ESCC" is based entirely on correlation. I therefore feel that, while likely, this is an overstatement of the actual data presented.

We agree with the reviewer that our data demonstrate that GRHL3 is required for APOBEC3A expression during differentiation as opposed to demonstrating a direct role for GRHL3 in APOBEC3A induction. We have revised the titles of Figure 4 (now "*Grainyhead-like transcription factor 3 is required for APOBEC3A expression during keratinocyte differentiation*") and the relevant results section (now "*APOBEC3A expression during keratinocyte differentiation requires Grainyhead-like transcription factor 3*" (line 229-230) accordingly. Similarly in cancer cells, our siRNA experiments in the BICR6 and BICR22 HNSCC cell lines demonstrate a requirement for GRHL3 in maintaining APOBEC3A expression. We have revised the subheading for this last Results section to "*Evidence for regulation of APOBEC3A by GRHL3 in SCC tissues and cell lines*" (line 304).

Comment 8: APOBEC3G is misspelled in line 368.

Corrected.

Comment 9: We did not have access to any supplementary tables for review.

We checked with the editorial team and as far as they could tell, our supplementary tables were available. We will ensure all reviewers have access to these when reviewing our revised submission.

Reviewer #2

Comment 1: Could the authors comment on the difference in scale between A3A and A3B expression in Fig 2e? It seems that A3B is expressed at much lower levels than A3A in tonsils. Is the same true in oesophagus (Fig S1)?

Revision Plan

Yes, this is correct. A3A is expressed at higher levels than A3B in both normal tonsil and oesophageal epithelium (at least at the mRNA level). This can be seen by comparing the A3A and A3B expression data in Figure 1 (the data displayed in Figure S1 is derived from the study by Madisson et al (2020), which represents 80,489 epithelial cells) and we have added a comment to the text (lines 108-110: “...and the mean *APOBEC3A* expression per cell was significantly higher than that of *APOBEC3B* in both tonsil and oesophagus” (Figure 1, $p < 2.22E-16$ (unpaired Wilcoxon’s Rank Sum test)).”

- There is a typo on line 239 where Figure 1e is called out, but I believe the authors meant 2e.

Corrected.

- Supplemental Fig 6d - the heatmap legend does not have color variation, which seems to be an error.

Corrected.

- In Fig 5b, was qPCR performed on sub-confluent cultures (as in Fig S9)?

Yes, BICR6 and BICR22 cells were sub-confluent at harvest in both cases. We have added this information to the legend for figure 5.

Reviewer #3

Regulation: Clarify if the link is unique to dysregulated cancer cells, or might reflect normal cellular processes amplified in oncogenesis.

Our initial identification of GRHL3 as a potential regulator of *APOBEC3A* was made using scRNA-seq data from healthy tonsil epithelium and validated using siRNA in NIKS – a spontaneously immortalised epidermal keratinocyte cell line, originating from neonatal foreskin. We also see a correlation between *APOBEC3A* expression and GRHL3 activity in healthy squamous oesophageal epithelium, so we propose that the link does indeed reflect normal cellular processes that could be amplified in oncogenesis, as the reviewer suggests. The same is true for the expression patterns of *APOBEC3B*.

Differentiation Focus: This study uniquely emphasises *APOBEC3A*'s potential in differentiation, an angle to explicitly highlight as it may have far-reaching implications beyond cancer understanding.

We agree that our observations do raise interesting questions regarding a possible role for *APOBEC3A* in differentiation (a point also noted by Reviewer 1). Some thoughts on this are included in the Discussion (lines 454-471) but in our revised manuscript we have also included mention in the abstract (lines 42-43), which now concludes “*These findings suggest that*

Revision Plan

APOBEC3A may play a functional role during keratinocyte differentiation and offer a mechanism for acquisition of APOBEC3A mutagenic activity in tumours". We are actively working on this question having deleted APOBEC3A in NIKS but would respectfully suggest (we think we are in agreement with the reviewer on this point) that the question of APOBEC3A's potential function(s) in differentiation lies beyond the scope of this manuscript.

4. Description of analyses that authors prefer not to carry out

Please include a point-by-point response explaining why some of the requested data or additional analyses might not be necessary or cannot be provided within the scope of a revision. This can be due to time or resource limitations or in case of disagreement about the necessity of such additional data given the scope of the study. Please leave empty if not applicable.

Reviewer #3 (Significance (Required)):

The link between GRHL3 and APOBEC3A discovered in the manuscript is compelling, however, further validation will strengthen the study. Directly manipulating GRHL3 in cancer models to observe changes in APOBEC-induced mutations would support the conclusion that it is a primary driver of this mechanism.

We agree that deletion of GRHL3 in cancer models (e.g. BICR6, BICR22) and assessment of the effects on APOBEC-mediated mutagenesis would provide strong evidence regarding the significance of this regulatory mechanism in cancer. Such experiments have been conducted for the deletion of *APOBEC3A* and *APOBEC3B* themselves (Petljak et al *Nature* 607(7920):799-807 (2022)), clearly demonstrating that loss of expression results in a loss of ongoing mutagenesis. These experiments are lengthy however (following the generation of suitable clones, they were cultured over several months to allow time for sufficient accumulation of mutations) and costly (whole genome sequencing of multiple clones is necessary to provide the statistical power required to observe the effect on mutagenesis). We suggest that in the interests of publishing our findings (which as all reviewers have pointed out, will be of considerable interest) in a timely manner, we include in our discussion the following sentence "while our data suggest aberrant activation of GRHL3-APOBEC3A signalling could be an important mechanism leading to APOBEC-mediated mutagenesis in cancer, this will need to be investigated further, for example by following the effect of GRHL3 deletion on APOBEC3A expression and the accumulation of APOBEC signature mutations in long-term cell culture studies. We note that APOBEC3A expression is only one part of the picture: mutagenesis also requires the availability of ssDNA substrate, for example during DNA replication. Again, our scRNA-seq analysis of a limited number of cancers suggests APOBEC3A is expressed in a minority of tumour cells that are undergoing DNA replication (and that these cells also exhibit GRHL3 activation) but whether this results in APOBEC-mediated mutagenesis requires further investigation."

Revision Plan

In lieu of demonstrating an effect on mutagenesis, we propose to do the following:

1. Demonstrating the effect of GRHL3 knockdown on APOBEC3A RNA editing activity in cancer cell lines. Measuring the extent of C>U editing at an APOBEC3A target site (C558) in the DDOST mRNA has been shown to be superior to APOBEC3A protein- and mRNA-based assays in predicting the activity of APOBEC3A on DNA (Jalili et al *Nature Communications* 11(1):2971 (2020)). It has been used either in parallel with (Petljak et al *Nature* 607(7920):799-807 (2022), or in place of (Isozaki et al *Nature* 620(7973):393-401 (2023)). We have shown in this manuscript (Figure 4e) that GRHL3 knockdown results in a decrease in DDOST mRNA editing and we propose to conduct this assay on the HNSCC cell lines (BICR6 and BICR22) in which we have observed a decrease in APOBEC3A mRNA levels following GRHL3 knockdown (Figure 5b).
2. Evaluating markers of replication stress (phospho-Chk1) and DNA damage (γ H2AX) that become elevated in cancer cells expressing APOBEC3A and that function as a more immediate biomarker for genotoxic activity than the eventual occurrence of mutations. We will conduct immunofluorescence microscopy and/or immunoblotting to monitor pChk1 and γ H2AX in NIKS and HNSCC cell lines +/- GRHL3 knockdown.
3. Examine whether tumours harbouring *GRHL3* deletions in The Cancer Genome Atlas database display reduced APOBEC signature mutation burdens.

Dear Dr. Fenton,

Thank you for transferring your manuscript with Review Commons referee reports and responses to The EMBO Journal.

Given the referees' positive recommendations, I would like to invite you to submit a revised version of the manuscript, addressing the comments of all three reviewers. I should add that it is EMBO Journal policy to allow only a single round of revision, and acceptance of your manuscript will therefore depend on the completeness of your responses in this revised version.

When preparing your letter of response to the referees' comments, please bear in mind that this will form part of the Review Process File, and will therefore be available online to the community. For more details on our Transparent Editorial Process, please visit our website.

Thank you for the opportunity to consider your work for publication. I look forward to your revision.

Yours sincerely,

Kelly M Anderson, PhD
Editor, The EMBO Journal
k.anderson@embojournal.org

We realize that it is difficult to revise to a specific deadline. In the interest of protecting the conceptual advance provided by the work, we recommend a revision within 3 months (20th Aug 2024). Please discuss the revision progress ahead of this time with the editor if you require more time to complete the revisions. Use the link below to submit your revision:

Link Not Available

Rev_Com_number: RC-2024-02419

New_manu_number: EMBOJ-2024-117861-T

Corr_author: Fenton

Title: Differentiation signals induce APOBEC3A expression via GRHL3 in squamous epithelia and squamous cell carcinoma

We thank the referees for their appraisal of our manuscript and are pleased to note their positive comments regarding the robustness of our data and the significance of our findings to a wide audience.

In addition to performing further experiments and analyses to address the specific comments below, we have also added the following:

- 1) We applied a method (inferCNV) to our scRNA-seq data from tumour samples to infer copy number variation (CNV) profiles for each cell by averaging gene expression levels over large genomic regions (see methods for details). This enabled us to identify 695 of 19,314 the epithelial cells from our tumour samples with inferred CNV profiles that closely resembled those of cells from our matched normal tonsil cells (Appendix Figure S8). These were classed as non-malignant and further analyses were based on the 18,619 malignant epithelial cells remaining. This has not altered any of the findings but allows us to state with confidence that when describing *APOBEC3A* and *APOBEC3B* expression patterns in tumours, we are indeed referring to malignant tumour cells and not to non-malignant cells within those samples. This is described in the Results text (lines 292-297) and in the Methods section.
- 2) We added a detailed description of the methods used to generate *APOBEC3A* KO NIKS that were used to demonstrate specificity of the anti-*APOBEC3A* antibody (see Methods section and Appendix Fig S14).

Revisions to the text are in blue font in the manuscript file and we've indicated the line numbers where revisions have been made.

Reviewer #1

Evidence, reproducibility and clarity

Summary

In this study, the authors performed single cell RNA sequencing to study the expression of *APOBEC3A* (A3A) and *APOBEC3B* (A3B) across different healthy and malignant tissue samples. The key finding of the study is that while A3B expression was restricted to proliferating cells, A3A was expressed mainly in differentiating normal cells. These results were validated by spatial transcriptomics, IHC and functional experiments. The study identified Grainyhead-like transcription factor 3 (GRHL3) as a transcriptional regulator of A3A in terminally differentiating keratinocytes and SCC tumor cells. GRHL3 was found to correlate with the expression and cytidine deaminase activity of A3A. Overall the study identifies specific cell lineages that express either A3A or A3B in normal tissues and identifies GRHL3 as a previously unknown transcriptional regulator of A3A in normal tissues and certain cancers.

Major comments-

1. The authors discuss breast cancers to be rich in A3B, consistent with cancer cell proliferation, however cancers like head and neck and that of the oesophagus appear to express both A3A and A3B. It is unclear from the description whether there are cases where A3A and A3B are

coexpressed in cancer. Can the authors determine this from their dataset?

We do indeed see a minority of A3A expressing cells that also express A3B, both in normal and cancer samples. We speculate that at least in normal epithelium, these are cells that may have exited cell cycle in G2/M and initiated differentiation, as has been described previously for keratinocytes. In cancers, co-expression may (also) occur due to dysregulation of either A3A or A3B expression, for example where we see A3A expression in a minority population of dividing cells in some HNSCC cases (Figure 5F). However, this population is too small to conduct a meaningful analysis (e.g. to determine which other genes are co-expressed / pathways enriched) in these A3A/A3B co-expressing cells. We have not been able to find any scRNA-seq datasets containing sufficient A3A/A3B co-expressing cells to determine whether there is anything different / unique about this population. This interesting question will have to be revisited once more scRNA-seq data are available from tissues that express both A3A and A3B.

2. In line 130 the authors state "To address this question, the 2,649 epithelial cells derived from normal tonsil were subset from our 21,937 epithelial cell dataset (Figure 2a) and pathway analysis using gene ontology biological processes (GOBP) was performed on the top 100 genes co-expressed with either APOBEC3A or APOBEC3B (Table S3)." This sentence is a bit confusing and looking at 2a, it is impossible to extrapolate what subset they are referring to. This statement and accompanying data could use some clarification for the reader.

We have clarified this section of the text to read as follows (lines 136-140): "To address this question, after further quality control (see Methods), we identified the top 100 genes co-expressed with either *APOBEC3A* or *APOBEC3B* (Table EV3) in the 2,649 epithelial cells from our healthy tonsil samples (Figure 2A, blue) and used pathway analysis to examine the gene ontology biological processes (GOBP) enriched among each geneset (Fig 2B)."

We have also amended the legend for Figure 2 as follows: "(a) UMAP projection of 22,595 epithelial cells from oropharyngeal squamous cell carcinoma samples (n = 10; 18,619 malignant cells (red), 695 non-malignant cells (purple), and matched normal tonsil (n = 7; 3,281 cells (blue)). b) GOBP terms enriched amongst the sets of 100 genes that were co-expressed with either *APOBEC3A* or *APOBEC3B* in the 2,649 (after QC) epithelial cells from normal tonsil."

3. In line 236-239 authors say, "Furthermore, GRHL3 was the only transcription factor among those whose activity was correlated with APOBEC3A expression that was significantly upregulated in the differentiating compartment of the normal tonsil epithelium, in which most APOBEC3A-expressing cells were clustered (Figure 4a, b, Figure 1e, Table S6)". However, there is no figure 1e provided.

The reference to Figure 1E was a mistake. We have corrected this to Figure 2E.

4. Was a similar peak motif also found in the promoters/enhancers of A3A for IVL and ELF3 that are also downregulated upon GRHL3 depletion?

We identified a GRHL3 binding peak containing three putative GRHL3 binding motifs in the *ELF3* promoter but not in the *IVL* promoter and have included a new supplementary figure (Appendix Fig S7B) and the following text to our Results section (lines 265-267) “GRHL3 and WDR5 binding to a region of the *ELF3* promoter containing three putative GRHL3 binding motifs was also evident in this ChIP-seq dataset (Appendix Fig S7) but no peaks were observed at the *IVL* promoter.” It should be noted that neither *ELF3* nor *IVL* have definitively been shown to be direct GRHL3 target genes but at least for *ELF3* (and *APOBEC3A*), the evidence from GRHL3 and WDR5 ChIP-seq is consistent with direct regulation).

From the data it would appear that A3A is transiently expressed during differentiation and then repressed in terminally differentiated cells, suggesting that some repressors must be subsequently recruited.

Thank you for raising this interesting point. We agree that this is certainly of interest to pursue in future studies, for example by using organotypic culture systems in which terminal differentiation can be achieved. We have added mention of this to the discussion (lines 511-517) “Repression of *APOBEC3B* expression by the E2F4/RB-containing DREAM (dimerization partner, RB-like, E2F and MuvB) complex has been reported (Periyasamy et al., 2017), which may explain its confinement to proliferating cells in healthy squamous epithelia. The absence of *APOBEC3A* in basal and terminally differentiated cells suggests it too is subject to transcriptional repression; another mode of regulation that if disrupted could enable bursts of mutagenic APOBEC activity in cancer cells.”

5. How specific is GRHL3 expression to keratinocytes? In other words, would this mechanism potentially be relevant in breast or other tissue that exhibits APOBEC mutational signatures?

To address this question, we used bulk RNA sequencing data from TCGA to analyse *APOBEC3A* and *GRHL3* mRNA expression and predicted GRHL3 transcriptional activity across cancers arising in different tissues and have included the results in a new figure (Fig EV2). Except for acute myeloid leukaemia (*APOBEC3A* expression is known to be high in myeloid cells but *GRHL3* is not, and our analysis of ChIP-seq and scRNA-seq data suggest differential regulation of *APOBEC3A* in monocytes and keratinocytes (Fig 4F, G), the cancer types that display the highest *APOBEC3A* expression are also those that express the most *GRHL3*, and the majority are squamous cell carcinomas.

While the strongest correlations between GRHL3 expression / activity and *APOBEC3A* expression are seen in oesophageal and cervical SCCs, we do see strong correlations also in adenocarcinomas from these tissues. *APOBEC3A* expression is significantly but weakly correlated with GRHL3 expression ($\rho = 0.19$, adj p = $4.07E-10$) and activity ($\rho = 0.17$, adj p = $2.0E-9$) in breast cancer (BRCA). We describe our analysis of TCGA RNA-seq data in the Results text (lines 317-341).

We also analysed a recently published single cell atlas of healthy human airways (nasopharynx to lungs) that clearly shows expression of *APOBEC3A* in secretory (differentiated) cells of the nose, which also express *GRHL3*, *IVL*, *KRT10* and *ELF3* (new Fig 4G), while *APOBEC3B* expression is mainly seen in ‘cycling basal’ cells, along with Ki-67 and MCM7 (new Fig EV1). These results further support our findings on *APOBEC3A* and *APOBEC3B* expression in distinct epithelial cell populations from healthy tonsil and

oesophagus and indicate higher *APOBEC3A* expression in the nose than in other parts of the airway epithelium. We describe these findings in the Results text (lines 269-277).

6. Did the authors identify any transcription factors in spatial scRNA-seq data that correlate with A3B expression?

We have added the A3B results of our SCENIC analysis to Table EV5 and have added the following to the results text (lines 280-285): “Although no single transcription factor activity displayed a consistently strong correlation with *APOBEC3B* expression (Table EV5), possibly due to the lower number of *APOBEC3B*-expressing cells in several of the datasets), correlations with E2F family transcription factors are consistent with previous reports linking repressive E2F-containing complexes to *APOBEC3B* regulation regulation (Periyasamy et al, 2017; Roelofs et al, 2020), and with its expression in proliferating cells.”

7. The results heading for Figure 4 states that A3A expression is induced by GRHL3. While the data is consistent with this, it is not actually demonstrated that expression of GRHL3 directly causes A3A induction. The converse is shown in depletion experiments, however I feel that stating A3A expression in differentiation requires GRHL3 or something to that effect more accurately reflects the data in the absence of evidence for a more cause-effect relationship. Similarly, the heading for the next section "GRHL3 drives APOBEC3A expression in HNSCC and ESCC" is based entirely on correlation. I therefore feel that, while likely, this is an overstatement of the actual data presented.

We agree that our data demonstrate that GRHL3 is required for APOBEC3A expression during differentiation as opposed to demonstrating a direct role for GRHL3 in APOBEC3A induction. We have revised the titles of Figure 4 (now “*Grainyhead-like transcription factor 3 is required for APOBEC3A expression during keratinocyte differentiation*”) and the relevant results section (now “*APOBEC3A expression during keratinocyte differentiation requires Grainyhead-like transcription factor 3*”) (line 213-214) accordingly. Similarly in cancer cells, our siRNA experiments in the BICR6 and BICR22 HNSCC cell lines demonstrate a requirement for GRHL3 in maintaining APOBEC3A expression. We have revised the subheading for this last Results section to “*Evidence for regulation of APOBEC3A by GRHL3 in SCC tissues and cell lines*” (line 287).

Minor comments

8. APOBEC3G is misspelled in line 368.

Corrected.

9. We did not have access to any supplementary tables for review.

We are sorry to hear this. We checked with the editorial team at *Review Commons*, who couldn't see any problems with the files and we hope these tables are now accessible.

Significance

General assessment: The study provides new information about the endogenous regulation of APOBECs, a topic for which there is little knowledge and considerable interest given their roles in cancer. It also adds additional detail to previous observations that EGFR inhibitors can activate A3A expression.

Advance: The authors demonstrate that A3A and A3B are differentially regulated and implicated GRHL3 as a main regulator of A3A in differentiating cells, also identifying a likely promoter element for A3A. Further, they provide evidence that A3B is expressed primarily in cycling cells. These observations raise interesting questions about the role of A3A in cancer. Many cancers exhibit a gradient of differentiation and these results may suggest that this process would lead to A3A expression and subsequent mutations that could lead to more aggressive tumor growth. Additionally, it suggests that there may be a role for A3A expression in differentiating cells under homeostatic conditions.

We thank the reviewer for highlighting the significance and implications of our study. The fact that our data suggest a potential role for A3A in differentiating keratinocytes was also raised by reviewer 3. Our discussion contains a comment on this (lines 464-466) and we have added mention in the abstract (lines 42-43).

Limitations of the study: The results rely heavily on the analysis of scRNAseq experiments. While these approaches are ideal for identifying cell populations, they are also not as sensitive as other methods for detecting the expression of genes. While some depletion and ChIP experiments are carried out, much of the model relies on correlation rather than direct manipulation of promoter elements etc. While all of the data is consistent with the author's proposals, it is a limitation of the study.

We agree that this is a limitation of the study, and that a detailed exploration of the mechanism by which GRHL3 regulates *APOBEC3A* expression will be an important next step. As noted in our response to comment 7, we have modified our wording to avoid overstating the evidence for direct regulation of *APOBEC3A* by GRHL3. We did conduct further analysis of ChIP-seq data for active chromatin marks (H3K27Ac, H3K4Me1 and H3K4Me3) upstream of the *APOBEC3A* transcriptional start site in differentiating keratinocytes versus monocytes – a cell type in which in which *APOBEC3A* is highly expressed but *GRHL3* is not (see for example, our analysis of the healthy human airways single cell atlas, new Figure 4G). Interestingly the -33kb enhancer, at which we see the GRHL3 and WDR5 binding peaks displays strong peaks for all three histone marks across multiple ChIP-seq datasets from keratinocytes but no peak is seen at this location in monocytes, which instead display abundant peaks at the -4kb and -15kb regulatory regions (new Figure 4F, new results text, lines 256-267). This suggests differential usage of *cis*-acting regulatory elements to drive *APOBEC3A* expression in keratinocytes and monocytes. The fact that we observe a GRHL3 binding peak in the -33kb enhancer lends support to a model for direct regulation but further work, beyond the scope of our current study, will be required to definitively demonstrate this.

Audience: The results will be of interest to a wide audience interested in cancer mutational signatures, cancer etiology, antiviral responses and DNA damage and repair.

We thank the reviewer for highlighting the broad audience to whom our manuscript will be of interest.

Expertise: Our expertise is in the mechanisms of the DNA damage response, including to APOBEC activity, and its impact on cancer etiology. We do not have significant experience with the generation or analysis of scRNAseq data.

Reviewer #2

Evidence, reproducibility and clarity

Summary: Smith and colleagues report novel findings related to regulation of expression of APOBEC3A and APOBEC3B, two candidate cancer mutagens, in both squamous cell cancers and healthy epithelia. Through de novo single cell RNA sequencing and analysis of published data sets, Smith et al find that the GRHL3 pioneer transcription factor regulates APOBEC3A to restrict expression to differentiating/terminally differentiated cells. In contrast, APOBEC3B is expressed in proliferating cells during G2-M phase and expression is associated with mitotic markers.

I have only minor comments/suggestions:

1. Could the authors comment on the difference in scale between A3A and A3B expression in Fig 2e? It seems that A3B is expressed at much lower levels than A3A in tonsils. Is the same true in oesophagus (Fig S1)?

Yes, this is correct. *APOBEC3A* is expressed at higher levels than *APOBEC3B* in both normal tonsil and oesophageal epithelium (at least at the mRNA level). This can be seen by comparing the *APOBEC3A* and *APOBEC3B* expression data in Figure 1 (the data displayed in Appendix Figure S1 is derived from the study by Madisson et al (2020), which represents 80,489 epithelial cells) and we have added a comment to the text (lines 119-121: “...and the mean *APOBEC3A* expression per cell was significantly higher than that of *APOBEC3B* in both tonsil and oesophagus” (Figure 1, $p < 2.22E-16$ (unpaired Wilcoxon’s Rank Sum test)).”

2. NIKS experiments in Fig 3/S2 would benefit from a non-qPCR marker of cell cycle and/or proliferation. While data regarding A3A/B expression correlated with proliferation/cycle markers are robust, it would be nice to see complementary experiments to show proliferation or cell cycle in this system.

We performed flow cytometry on fixed, propidium iodide-stained cells collected under these conditions to provide a complementary and more direct assessment of cell cycle status in these experiments. These results have been added to Figure 3 and to Appendix Figure S2.

3. Does GRHL3 mediate A3A expression in non-afatinib induction conditions? In other words, if

one of the other systems used in Fig 3 were subject to GRHL3 depletion, would A3A expression be inhibited?

We have added new data showing that *GRHL3* siRNA reduces the level of *APOBEC3A* expression seen following acute (3-hour) treatment of NIKS with the known *APOBEC3A* inducer, 12-myristate 13-acetate (PMA, Appendix Figure S6A), and following withdrawal of all growth factors (Appendix Fig S6B) or following removal of EGF alone (Appendix Fig 6C). We have added the following to the results text (lines 241-243): “*GRHL3* knockdown also decreased *APOBEC3A* mRNA levels in PMA-treated NIKS (Appendix Fig S6A), and following growth factor withdrawal (Appendix Fig S6B, C).”

We were not able to examine the effect of *GRHL3* siRNA upon induction of *APOBEC3A* at high cell density due to the technical difficulty of achieving sufficient transfection efficiency followed by sufficient cell proliferation to achieve the cell density at which we see *APOBEC3A* induction (indeed, we were not able to achieve sufficient cell densities to observe optimal induction of A3A in response to PMA or growth factor withdrawal following transfection, as noted in the legend to Appendix Fig S6 but we were able to achieve sufficient induction to be able to observe the effects of *GRHL3* siRNA). Given the consistency with which we observe the effects of *GRHL3* siRNA on *APOBEC3A* expression in response to afatinib, starvation and PMA however (in addition to the scRNA-seq data showing *APOBEC3A* is expressed in differentiating keratinocytes in human tissue), we fully expect that *GRHL3* is also required for *APOBEC3A* expression in density-induced differentiation.

4. In Fig 4f the *GRHL3* transcription factor is identified from in silico data at two regulatory sites proximal to A3A. *WDR5* is presumably recruited by *GRHL3* and is found at similar locations. Do the authors think that the same chromatin regulation of A3A occurs in all cells?

We thank the referee for raising this interesting point. Apart from keratinocytes, *APOBEC3A* is also highly expressed in monocytes/macrophages. Based on their expression patterns, it is unlikely that *GRHL3* or other Grainyhead family transcription factors are involved in regulating *APOBEC3A* in myeloid cells (indeed, there is no *GRHL3* expression in monocytes or macrophages from the single cell atlas of healthy human airways that we analysed during these revisions but *APOBEC3A* is expressed (new Figure 4G).

We therefore compared ChIP-seq data for active histone marks (H3K27Ac, H3K4Me1 and H3K4Me3) from keratinocytes and monocytes. Interestingly we see clearly distinct patterns, with minimal peaks at the -33kb region in monocytes (in which much stronger peaks are seen at predicted regulatory regions -15kb and -4kb from the *APOBEC3A* TSS). In keratinocytes, the histone modification peaks are strongest at the -33kb region where we see evidence of *GRHL3* and *WDR5* binding. So it would appear that *APOBEC3A* is regulated by different *cis*-acting elements in keratinocytes and monocytes. We have added these important new findings in a new version of Figure 4F and have described them in the text (lines 256-267).

5. NFκb has been reported as a transcriptional mediator of *APOBEC3A* expression in two recent publications (Isozaki, et al PMID 37407818, Oh, et al PMID 34389714). However, NFκb does not appear in the list of transcription factors whose activity correlated with *APOBEC3A*

expression in keratinocytes (Fig 4a). Can the authors comment on where NFkB might fit into the GRHL3-APOBEC3A axis?

As the reviewer points out, we do not see a positive correlation between inferred RELA activity and *APOBEC3A* expression in scRNA-seq data, at least from normal tonsil. We note that in the study by Oh et al, RELA is required for *APOBEC3A* induction by replication stress in cancer cell lines, while in the study by Isozaki et al, NFkB1 was found to mediate expression in *EGFR*-mutant lung cancer cell lines treated with EGFR inhibitor. Although we also see induction of *APOBEC3A* in response to EGFR inhibition in keratinocytes, the context is distinct. In the EGFR-mutant ('EGFR-addicted') NSCLC cell lines used by Isozaki et al, EGFR inhibition causes apoptosis in most cells, leaving a subpopulation of what they describe as drug tolerant persister cells, in which NFkB-dependent *APOBEC3A* induction is observed.

EGFR inhibition in normal keratinocytes does not cause cell death (at least over the 24-hour treatments used) but instead drives differentiation, so the cellular response is quite different. To check whether GRHL3 may play a role in *APOBEC3A* induction in drug tolerant persister cells, we transfected PC9 cells (as used by Isozaki et al) with GRHL3 siRNAs but did not see a dependence on GRHL3 for *APOBEC3A* induction in response to EGFR inhibition in this EGFR-mutant lung cancer cell line, consistent with a distinct mechanism acting in this scenario.

Given NFkB's established role as a mediator of responses to a variety of cellular stresses, we hypothesize that *APOBEC3A* induction by replication stress (Oh et al) or in *EGFR*-mutant drug-tolerant persister cells that survive EGFR inhibition (Isozaki et al) are both examples of NFkB-mediated stress responses, while *APOBEC3A* induction during differentiation appears to be a normal physiological process (as evidenced by our data from healthy tonsil, oesophagus and airways).

We do see positive correlations between predicted NFkB activity and *APOBEC3A* expression in some of the scRNA-seq datasets from tumours (particularly so in oesophageal SCC, where it is possible that replication stress and/other stresses encountered by tumour cells may act to induce *APOBEC3A* via NFkB). GRHL3 has been described as a pioneer factor (Jacobs et al *Nat. Genet.* 50, 1011–1020 (2018); a transcription factor that can bind to regions of closed chromatin and recruit chromatin remodellers required for the initial priming of a gene for activation by additional transcription factors. We hypothesize that GRHL3 may therefore be necessary for the activation of *APOBEC3A* expression by other transcription factors (potentially including NFkB), at least in keratinocytes. We plan to include experiments to test this as part of a detailed analysis of how GRHL3 regulates *APOBEC3A* but this goes beyond the scope of the present manuscript.

Minor clarifications:

- There is a typo on line 239 where Figure 1e is called out, but I believe the authors meant 2e.

Corrected.

- Supplemental Fig 6d - the heatmap legend does not have color variation, which seems to be an error.

Corrected.

- In Fig 5b, was qPCR performed on sub-confluent cultures (as in Fig S9)?

Yes, BICR6 and BICR22 cells were sub-confluent at harvest in both cases. We have added this information to the legend for figure 5 and it is also in the Methods.

Significance

This manuscript tackles an essential problem in cancer mutagenesis, which is the normal and aberrant regulation of the APOBEC enzymes. The data are robust and the manuscript is well written. The switch between A3B expression in cycling cells and A3A expression in differentiating cells fills a gap in knowledge that makes sense of many prior published findings. The impact of this manuscript is substantial and sets up the field for further analysis of APOBEC regulation and dysregulation in both healthy and malignant tissues beyond keratinocytes and squamous cell tumors.

We thank the reviewer for highlighting the substantial impact of our manuscript and the robustness of our data.

Reviewer #3 (Evidence, reproducibility and clarity (Required)):

This study explores APOBEC3A and APOBEC3B expression, regulation, and their implications for normal cellular function versus potential cancer-related dysregulation. The emphasis is on squamous cell carcinomas of head and neck (HNSCC) and esophagus (ESCC).

Approach: A robust, multi-pronged approach is used, including single-cell RNA sequencing (scRNA-Seq), spatial transcriptomics, antibody validation, and computational analysis.

Findings:

- GRHL3 emerges as a central regulator of APOBEC3A, particularly in keratinocytes. APOBEC3B expression aligns with cell division patterns, supporting its role in DNA replication-linked mutagenicity.
- The correlation of GRHL3, APOBEC3A, and cellular replication status in HNSCC and ESCC suggests a potential oncogenic mechanism.

Conclusions:

APOBEC3A appears to have a tightly controlled role in normal skin cell differentiation with implications for skin health. GRHL3, or its associated signaling networks, may represent promising therapeutic targets in cancers driven by APOBEC3-related mutations.

Reviewer #3 (Significance (Required)):

The link between GRHL3 and APOBEC3A discovered in the manuscript is compelling, however, further validation will strengthen the study. Directly manipulating GRHL3 in cancer models to

observe changes in APOBEC-induced mutations would support the conclusion that it is a primary driver of this mechanism.

We agree that deletion of GRHL3 in cancer models (e.g. BICR6, BICR22) and assessment of the effects on APOBEC-mediated mutagenesis would provide strong evidence regarding the significance of this regulatory mechanism in cancer. Such experiments have been conducted for the deletion of *APOBEC3A* and *APOBEC3B* themselves (Petljak et al *Nature* 607(7920):799-807 (2022)), clearly demonstrating that loss of expression results in a loss of ongoing mutagenesis. These experiments are lengthy however (following the generation of suitable clones, they were cultured over several months to allow time for sufficient accumulation of mutations) and costly (whole genome sequencing of multiple clones is necessary to provide the statistical power required to observe the effect on mutagenesis). We suggest that in the interests of publishing our findings (which as all reviewers have pointed out, will be of considerable interest) in a timely manner, we include in our discussion the following sentence “while our data suggest aberrant activation of GRHL3-APOBEC3A signalling could be an important mechanism leading to APOBEC-mediated mutagenesis in cancer, this will need to be investigated further, for example by following the effect of GRHL3 deletion on *APOBEC3A* expression and the accumulation of APOBEC signature mutations in long-term cell culture studies. We note that *APOBEC3A* expression is only one part of the picture: mutagenesis also requires the availability of ssDNA substrate, for example during DNA replication. Again, our scRNA-seq analysis of a limited number of cancers suggests *APOBEC3A* is expressed in a minority of tumour cells that are undergoing DNA replication (and that these cells also exhibit GRHL3 activation) but whether this results in APOBEC-mediated mutagenesis requires further investigation.”

We did assess markers of acute genotoxicity (gamma-H2AX and phospho-CHK1) by immunofluorescence microscopy following 24h treatment of WT or *APOBEC3A* KO NIKS to afatinib but saw no increase in these markers, and no difference between WT and *APOBEC3A* KO clones.

GRHL3 Specificity:

Tissue Context: It's important to determine if GRHL3-mediated APOBEC3A activation is unique to cancers explored here, or a broader principle in epithelial malignancies.

We agree that it is important to understand how widespread this mechanism of APOBEC3A regulation may be. As detailed in our response to Reviewer #1, comment 5:

To address this question, we used bulk RNA sequencing data from TCGA to analyse *APOBEC3A* and *GRHL3* mRNA expression and predicted GRHL3 transcriptional activity across cancers arising in different tissues and have included the results in a new figure (Fig EV2). Except for acute myeloid leukaemia (*APOBEC3A* expression is known to be high in myeloid cells but *GRHL3* is not, and our analysis of ChIP-seq and scRNA-seq data suggest differential regulation of *APOBEC3A* in monocytes and keratinocytes (Fig 4F, G)), the cancer types that display the highest *APOBEC3A* expression are also those that express the most *GRHL3*, and the majority are squamous cell carcinomas.

While the strongest correlations between GRHL3 expression / activity and *APOBEC3A* expression are seen in oesophageal and cervical SCCs, we do see strong correlations also in adenocarcinomas from these tissues. *APOBEC3A* expression is significantly but

weakly correlated with GRHL3 expression ($\rho = 0.19$, adj $p = 4.07E-10$) and activity ($\rho = 0.17$, adj $p = 2.0E-9$) in breast cancer (BRCA). These analyses are described in the Results text, lines 317-341.

We also analysed a recently published single cell atlas of healthy human airways (nasopharynx to lungs) that clearly shows expression of *APOBEC3A* in secretory (differentiated) cells of the nose, which also express *GRHL3*, *IVL*, *KRT10* and *ELF3* (new Fig 4G), while *APOBEC3B* expression is mainly seen in 'cycling basal' cells, along with *MKI67* and *TOP2A* (new Fig EV1). These results further support our findings on *APOBEC3A* and *APOBEC3B* expression in distinct epithelial cell populations from healthy tonsil and oesophagus and indicate higher *APOBEC3A* expression in the nose than in other parts of the airway epithelium. These findings are described in the Results text, lines 269-277.

Regulation: Clarify if the link is unique to dysregulated cancer cells, or might reflect normal cellular processes amplified in oncogenesis.

Our initial identification of GRHL3 as a potential regulator of *APOBEC3A* was made using scRNA-seq data from healthy tonsil epithelium and validated using siRNA in NIKS – a spontaneously immortalised epidermal keratinocyte cell line, originating from neonatal foreskin. We also see a correlation between *APOBEC3A* expression and GRHL3 activity in healthy squamous oesophageal epithelium, so we propose that the link does indeed reflect normal cellular processes that could be amplified in oncogenesis, as the reviewer suggests. The same is true for the expression patterns of *APOBEC3B*.

Differentiation Focus: This study uniquely emphasises *APOBEC3A*'s potential in differentiation, an angle to explicitly highlight as it may have far-reaching implications beyond cancer understanding.

We agree that our observations do raise interesting questions regarding a possible role for *APOBEC3A* in differentiation (a point also noted by Reviewer 1). This is noted in our Discussion (lines 464-466), and in our revised manuscript we have also included mention in the abstract (lines 42-43), which now concludes "*These findings suggest that APOBEC3A may play a functional role during keratinocyte differentiation and offer a mechanism for acquisition of APOBEC3A mutagenic activity in tumours*". We are actively working on this question having deleted *APOBEC3A* in NIKS but would respectfully suggest (we think we are in agreement with the reviewer on this point) that the question of *APOBEC3A*'s potential function(s) in differentiation lies beyond the scope of this manuscript.

Therapeutic Breadth: Explore if the results are likely limited to the studied cancers, or have potential application across a wider range of epithelial cancers.

The findings we made in response to the point concerning GRHL3 specificity (tissue context, and to Reviewer #1, comment 5 indicate that *APOBEC3A* and *GRHL3* are co-expressed at highest levels in squamous cell carcinomas of the head/neck, cervix oesophagus and lung (Fig EV2). We do see strong correlations between *GRHL3* and

APOBEC3A expression also in adenocarcinomas of the cervix and oesophagus, albeit the levels of expression are lower in these tumours. Weaker but significant correlations are also observed in breast cancer and bladder cancer, suggesting this mechanism may operate across a wide range of epithelial cancers but this will need to be further investigated.

This study presents a significant contribution to our understanding of APOBEC3A and APOBEC3B function, with particular emphasis on their roles in both normal cellular processes and cancer development. The methodological rigor, including the multi-pronged approach and focus on cellular context, provides a detailed and insightful view. The identification of GRHL3 as a central regulator of APOBEC3A presents a promising novel therapeutic target for combating APOBEC-related mutations in cancer. Furthermore, by highlighting APOBEC3A's potential role in cellular differentiation, the study offers a new perspective on these enzymes that could have broader implications for understanding skin health and other physiological processes. While further experimental validation would definitively solidify the causative link between GRHL3 and APOBEC-induced mutagenesis, this work meaningfully advances the field. Its conceptual insights into episodic mutation and potential dysregulation of differentiation in cancer have the potential to influence future research directions and cancer treatment strategies.

We thank the reviewer for highlighting the significance and methodological rigour of our study, and for its potential to influence future research and cancer treatment strategies.

Dear Dr Fenton,

Thank you for submitting your revised manuscript (EMBOJ-2024-117861R) to The EMBO Journal. Please note that I have taken over this process from my editorial colleague Kelly Anderson since she recently progressed to a different role outside the office. Your amended study was sent back to the three referees for their scientific re-evaluation, and we have received detailed comments from two of them, which I enclose below. As you will see, the experts state that the work has been substantially improved by the revisions and they are now broadly in favour of publication. Please note that we have editorially assessed your response to referee #3 and found the concerns to be addressed satisfactorily.

Thus, we are pleased to inform you that your manuscript has been accepted in principle for publication in The EMBO Journal.

We now need you to take care of a number of issues related to formatting and data presentation as detailed below, which should be addressed at re-submission.

Please contact me at any time if you have additional questions related to below points.

As you might have seen on our web page, every paper at the EMBO Journal now includes a 'Synopsis', displayed on the html and freely accessible to all readers. The synopsis includes a 'model' figure as well as 2-5 one-short-sentence bullet points that summarize the article. I would appreciate if you could provide this figure and the bullet points.

Thank you for giving us the chance to consider your manuscript for The EMBO Journal. I look forward to your final revision.

Again, please contact me at any time if you need any help or have further questions.

Best regards,

Daniel Klimmeck

>> Recheck functionality of the author e-mail address for co-author Z.K. .

>> Limit the number of keywords for your study to maximally five.

>> Author Contributions: Please remove the author contributions information from the manuscript text. Note that CRediT has replaced the traditional author contributions section as of now because it offers a systematic machine-readable author contributions format that allows for more effective research assessment. and use the free text boxes beneath each contributing author's name to add specific details on the author's contribution.

More information is available in our guide to authors.
<https://www.embopress.org/page/journal/14602075/authorguide>

>> Dataset EV legends: The legends for all EV tables do not need to be duplicated and zipped with the excel files if they are already present in the files themselves. Please rename Tables EV 4,5,7,8,9 and 10 to make them Dataset EV1 - 6. Please adjust the numbering of the remaining EV tables accordingly, and update the callouts in the manuscript text.

>> Appendix file: the appendix file needs a ToC added on its first page, including page numbers.

>> Add a separate 'Statistical Analysis' section to the Methods part, detailing the algorithms and statistical tests applied.

>> References: adjust reference format to EMBO Journal format, 10 authors et al, and remove the DOIs.

>> Callouts: There is a callout for a Supp Table 2 in the legend of Fig 1, please correct; add a callout for Fig 3H in the running text.

>> Consider additional changes and comments from our production team as indicated below:

- Figure legends:

1. Please note that the legend for figure 3g-h is mislabeled as figure 3b, d in the statistical test section of the legend in the manuscript. This needs to be rectified.
2. Please define the annotated p values ***/**/* as well as provide the exact p-values for the same in the legend of figure EV 3a; as appropriate.
3. Please note that the exact p values are not provided in the legends of figures 2e; 3a, c, e, g-h; 4c-e; 5b.
4. Please indicate the statistical test used for data analysis in the legends of figures 2b; 4a; EV 3a.
5. Please note that the box plots need to be defined in terms of minima, maxima, centre, bounds of box and whiskers, and percentile in the legends of figures 4c; 5f; EV 2a-c.
6. Please note that information related to n is missing in the legends of figures 2e; 4c; 5f; EV 2a-b.

Please click on the link below to submit the revision:

Link Not Available

Referee #1:

Summary

The authors performed single cell RNA sequencing to study the expression of APOBEC3A (A3A) and APOBEC3B (A3B) across different healthy and malignant tissue samples. The key finding of the study is that while A3B expression was restricted to proliferating cells, A3A was expressed mainly in differentiating normal cells. These results were validated by spatial transcriptomics, IHC and functional experiments. The study identified Grainyhead-like transcription factor 3 (GRHL3) as a transcriptional regulator of A3A in terminally differentiating keratinocytes and SCC tumor cells. GRHL3 was found to correlate with the expression and cytidine deaminase activity of A3A. Overall the study identifies specific cell lineages that express either A3A or A3B in normal tissues and identifies GRHL3 as a previously unknown transcriptional regulator of A3A in normal tissues and certain cancers. These data suggest that A3A may have roles during normal differentiation and help to identify regulatory networks that may be altered in A3A expressing cancers that are identifiable due to their distinct mutagenic signature.

Major concerns

The authors have adequately addressed all major concerns that included several typos, questions about the applicability of the data to other cancers and the strength of some conclusions by including additional data and analysis, text corrections and additional discussion. They have also, in my opinion, thoughtfully addressed many of the concerns raised by other reviewers. I do not have any additional major or minor concerns and feel the study is suitable for publication.

Referee #2:

The authors have adequately addressed all of my queries and, in my opinion, those of other referees. This manuscript is suitable for publication.

Rev_Com_number: RC-2024-02419

New_manu_number: EMBOJ-2024-117861R

Corr_author: Fenton

Title: Differentiation signals induce APOBEC3A via GRHL3 in squamous epithelia and squamous cell carcinoma

The authors addressed the remaining editorial issues.

Dear Dr Fenton,

Thank you for submitting the revised version of your manuscript EMBOJ-2024-117861R1. I have now evaluated your amended manuscript and concluded that the remaining minor concerns have been sufficiently addressed.

I am thus pleased to inform you that your manuscript has been accepted for publication in the EMBO Journal.

On a different note, I would like to alert you that EMBO Press offers a format for a video-synopsis of work published with us, which essentially is a short, author-generated film explaining the core findings in hand drawings, and, as we believe, can be very useful to increase visibility of the work. Please see the following link for representative examples and their integration into the article web page:

<https://www.embopress.org/doi/full/10.15252/emj.2019103932>

Best regards,

Daniel Klimmeck

Daniel Klimmeck, PhD
Senior Editor
The EMBO Journal
EMBO
Postfach 1022-40
Meyerohofstrasse 1
D-69117 Heidelberg
contact@embojournal.org
Submit at: <http://emboj.msubmit.net>

Rev_Com_number: RC-2024-02419

New_manu_number: EMBOJ-2024-117861R1

Corr_author: Fenton

Title: Differentiation signals induce APOBEC3A via GRHL3 in squamous epithelia and squamous cell carcinoma